# Spatial heterogeneity of bone marrow endothelial cells unveils a distinct subtype in the epiphysis

Takahito Iga[1,2], Hiroshi Kobayashi[3], Dai Kusumoto[4], Tsukasa Sanosaka [5], Nobuyuki Fujita[6], Ikue Tai-Nagara[1], Tomofumi Ando[1,7], Tomoko Takahashi[1], Koichi Matsuo [8], Katsuto Hozumi [9], Kosei Ito [10], Masatsugu Ema [11], Takeshi Miyamoto[12], Morio Matsumoto[2], Masaya Nakamura[2], Hideyuki Okano [5], Shinsuke Shibata[13], Jun Kohyama[5], Kevin K. Kim[14], Keiyo Takubo [3] ✉ & Yoshiaki Kubota [1] ✉

Bone marrow endothelial cells (BMECs) play a key role in bone formation and haematopoiesis. Although recent studies uncovered the cellular taxonomy of stromal compartments in the bone marrow (BM), the complexity of BMECs is not fully characterized. In the present study, using single-cell RNA sequencing, we defined a spatial heterogeneity of BMECs and identified a capillary subtype, termed type S (secondary ossification) endothelial cells (ECs), exclusively existing in the epiphysis. Type S ECs possessed unique phenotypic characteristics in terms of structure, plasticity and gene expression profiles. Genetic experiments showed that type S ECs atypically contributed to the acquisition of bone strength by secreting type I collagen, the most abundant bone matrix component. Moreover, these cells formed a distinct reservoir for haematopoietic stem cells. These findings provide the landscape for the cellular architecture in the BM vasculature and underscore the importance of epiphyseal ECs during bone and haematopoietic development.

Blood vessels are necessary for the development and maintenance of organs throughout our bodies. They provide oxygen and nutrients to parenchymal cells in each tissue and secrete paracrine molecules, known as angiocrine factors, which drive cell differentiation and tissue morphogenesis[1–3]. This vascular function is mediated by organ-specific endothelial cells (ECs), the cellular and molecular heterogeneity of which have been intensively studied using single-cell transcriptomic technologies[4,5].

[1]Department of Anatomy, Keio University School of Medicine, Tokyo, Japan. [2]Department of Orthopedic Surgery, Keio University School of Medicine, Tokyo, Japan. [3]Department of Stem Cell Biology, Research Institute, National Center for Global Health and Medicine, Tokyo, Japan. [4]Department of Cardiology, Keio University School of Medicine, Tokyo, Japan. [5]Department of Physiology, Keio University School of Medicine, Tokyo, Japan. [6]Department of Orthopaedic Surgery, School of Medicine, Fujita Health University, Aichi, Japan. [7]Department of Surgery, Keio University School of Medicine, Tokyo, Japan. [8]Laboratory of Cell and Tissue Biology, Keio University School of Medicine, Tokyo, Japan. [9]Department of Immunology, Tokai University School of Medicine, Kanawaga, Japan. [10]Department of Molecular Bone Biology, Graduate School of Biomedical Sciences, Nagasaki University, Nagasaki, Japan. [11]Department of Stem Cells and Human Disease Models, Research Center for Animal Life Science, Shiga University of Medical Science, Shiga, Japan. [12]Department of Orthopedic Surgery, Kumamoto University, Kumamoto, Japan. [13]Division of Microscopic Anatomy, Graduate School of Medical and Dental Sciences, Niigata University, Niigata, Japan. [14]Division of Pulmonary and Critical Care Medicine, Department of Internal Medicine, University of Michigan, Ann Arbor, MI, USA. ✉e-mail: keiyot@gmail.com; ykubo33@a3.keio.jp

Bone marrow endothelial cells (BMECs) play a central role in the spatiotemporal relationship between angiogenesis and osteogenesis, referred to as angiogenic–osteogenic coupling[6–10]. They also serve as a stem cell niche component that supports the self-renewal and differentiation of haematopoietic stem and progenitor cells (HSPCs)[11–14]. Recent studies utilizing single-cell RNA sequencing (scRNA-seq) uncovered the cellular taxonomy of stromal compartments in the bone marrow (BM)[15,16], which enhanced our understanding of cell-to-cell interactions governing bone metabolism as well as life-long haematopoiesis. However, information about the spatiotemporal and functional composition of BMECs is limited and this must be analysed in more detail.

In the present study, we profiled BMECs isolated from mouse femurs using scRNA-seq and defined their heterogeneity. The data uncovered an unrecognized capillary subtype, termed type S (secondary ossification) ECs, exclusively existing in the epiphysis, the end of long bones[17,18]. Type S ECs possess unique phenotypic characteristics, which contributed to the maintenance of bone strength and constituted a distinct HSPC niche. Our findings provided the landscape for the cellular architecture in the BM vasculature and enhanced our understanding of the fundamental mechanism regulating osteogenesis and haematopoiesis.

## Results

### Spatial heterogeneity of transcriptomes in BMECs

As an initial step to explore the heterogeneity of BMECs, we conducted droplet-based scRNA-seq of CD31⁺CD45⁻ BM cells isolated from three surgically separated fragments of femurs, diaphyses, metaphyses and epiphyses of postnatal day 11 (P11) mice (Fig. 1a and Extended Data Fig. 1a–c). Using the Uniform Manifold Approximation and Projection (UMAP) of cells, we defined 11 EC clusters, including arterial, venous and capillary ECs, with few contaminations of osteoblasts (OBs)/mesenchymal stromal cells (MSCs)[15,19], erythroid cells and myeloid cells (Fig. 1b–d) based on known markers[4,5,20,21]. Type H and type L ECs have been histologically characterized as specialized capillary populations predominantly existing in the metaphysis and diaphysis, respectively[22,23]. For capillary EC clusters, based on the spatial origin and known markers[14,24], we assigned one cluster to type H ECs and five clusters to type L ECs (Fig. 1b–e). Type L ECs were divided into sinusoidal (SEC) and non-sinusoidal (non-SEC) ECs based on known SEC markers such as *Stab2* and *Vcam1*. Although transcription of *Pecam1* and *Emcn*, which are immunohistochemical markers of type H ECs, did not strikingly differ among capillary subtypes, expression of type H markers such as *Efnb2* and *Sox17* (ref. 14) was highest in the capillary cluster annotated as type H ECs (Fig. 1e). Intriguingly, we identified a unique non-SEC capillary population, distinct from any other ECs, which exclusively originated from the epiphysis. The development of this area is called the secondary ossification centre (SOC), and is the reason why we named this EC population type S ECs (Fig. 1c, cluster 8). By extracting genes upregulated in type S ECs compared with all EC clusters, *Ly6a* and *Ly6c1*, which are highly expressed in arterial ECs (AECs)[16] and type H ECs[14], were the no. 1 and no. 8 ranked upregulated genes in terms of the average log(fold-change) (log(FC)) (Fig. 1f and Supplementary Table 1). Intriguingly, *Col1a1* and *Col1a2*, both of which encode type I collagen, were also upregulated in type S ECs (Fig. 1f,g, Extended Data Fig. 1d and Supplementary Table 1), although feature plots of these genes showed that they were most highly expressed in OBs/MSCs (Fig. 1g). This atypical expression of type I collagen genes suggested that type S ECs play a distinct angiocrine role during osteogenesis. The other collagen genes were diversely expressed in type S, type H and type L ECs (Fig. 1h). Ingenuity Pathway Analysis (IPA) uncovered the top four networks represented by differentially expressed genes in type S ECs, including growth factor signalling (network 2) and integrin–extracellular matrix interaction (network 4) (Extended Data Fig. 1e). Flow cytometric analysis of isolated CD31⁺CD45⁻ BM cells confirmed that type S

ECs were characterized by Ly6a^high^Ly6c^high^ expression on their surface (Fig. 1i). Immunohistochemistry confirmed that Ly6a expression was higher in type S ECs than in type H ECs (Fig. 1j). These data demonstrate that type S ECs possess unique phenotypic characteristics, although some of them overlap with those of type H ECs and AECs.

### Molecular and morphological features of type S ECs

Recent advances in histological techniques enabled clear visualization of angiogenic–osteogenic coupling, focusing on the metaphysis[22,23]. Considering the scRNA-seq data described above and the existence of a unique niche maintaining skeletal stem cells in the epiphysis[17,18], which suggests that there is a differential mode of angiogenic–osteogenic coupling in this area, we thoroughly examined the spatiotemporal transition of angiogenesis and osteogenesis during postnatal development from P7 to P28 (Fig. 2a–l). ECs first invaded the epiphysis at P7 from two entry points on the anterior side (Fig. 2a,h,j,m and Supplementary Video 1) and rapidly expanded their network until weaning (Fig. 2c–g). Ossification proceeded synchronously (Fig. 2a–g). The overall shape of blood vessels in this area was tree like or dendritic, in contrast with the columnar morphology of those vessels in the metaphysis (Fig. 2h–l,n). The growing edge of type S vessels highly expressed CD31 and projected short filopodia towards avascular areas such as type H vessels (Fig. 2o and Extended Data Fig. 2a). Expression of Vegfr1 strikingly differed between type H and type S cells: type H ECs highly expressed Vegfr1, whereas Vegfr1 expression was barely detectable in type S ECs in *Vegfr1-BAC-RFP* mice (Fig. 2o and Extended Data Fig. 2b). Skeletal growth in mice is largely completed by 10 weeks after birth. Therefore, we examined Vegfr1 expression and found that it was retained even at 6 months of age (Fig. 2p). Expression of other vascular endothelial growth factor (VEGF) receptors, Vegfr2 and Vegfr3, did not apparently differ between type S and type H ECs (Fig. 2o and Extended Data Fig. 2c,d); these receptors were both highly expressed in type L ECs and were strikingly expressed in the growing edges of type S and type H ECs.

### Type S vessels are crucial for epiphyseal osteogenesis

Considering the lack of Vegfr1, a negative regulator buffering VEGF signalling[1], we suspected that type S ECs were more vulnerable to deletion of *Vegfr2* than type H ECs. Endothelial-specific *Vegfr2*-knockout (*Cdh5-BAC-Cre^ERT2+^Vegfr2^flox/flox^*, hereafter referred to as *Vegfr2^iΔEC^*) mice showed severely impaired angiogenesis and osteogenesis in the epiphysis during the early postnatal weeks (Fig. 3a–d and Extended Data Fig. 2e). In agreement with a previous report[25], the morphology of type H vessels was abnormal in *Vegfr2^iΔEC^* mice, but the impairment was milder than that of type S vessels (Fig. 3b and Extended Data Fig. 2f). Accordingly, metaphyseal ossification was not apparently delayed in *Vegfr2^iΔEC^* mice (Fig. 3c,d). The numbers of Runx2⁺ pre-osteoprogenitors[26] and osterix (Osx)⁺ osteoprogenitors[27], which usually exist around blood vessels[10], were significantly reduced in the epiphysis of *Vegfr2^iΔEC^* mice (Fig. 3e–k). Mice with endothelial-specific deletion of Dll4 (*Cdh5-BAC-Cre^ERT2+^Dll4^flox/flox^*, hereafter referred to as *Dll4^iΔEC^*), the predominant Notch ligand in ECs, showed a dense but reduced vascularized area in the epiphysis (Fig. 3l,m and Extended Data Fig. 2e). Accordingly, *Dll4^iΔEC^* mice showed reduced osteogenesis as well as decreased numbers of pre-osteoprogenitors and osteoprogenitors (Fig. 3n–v). Notch signalling in osteogenic progenitors is reportedly critical for bone fracture repair[28]; therefore, we deleted *Rbpj*, the principal effector of Notch signalling, in Osx-expressing mesenchymal progenitors (*Osx–Cre^ERT2+^Rbpj^flox/flox^*, hereafter referred to as *Rbpj^iΔosx^*). However, *Rbpj^iΔosx^* mice had no apparent defect in osteogenesis (Extended Data Fig. 3a–h). In retinal angiogenesis, *Dll4* deletion was reported to overcome the anti-angiogenesis effects of *Vegfr2* deletion[29]. By contrast, in vascularization of the epiphysis, endothelial ablation of *Vegfr2* prevented the hypervascularity induced by *Dll4* deletion (Extended Data Fig. 3i–m). *Csf1^op/op^* mutant mice, in which osteoclasts are almost

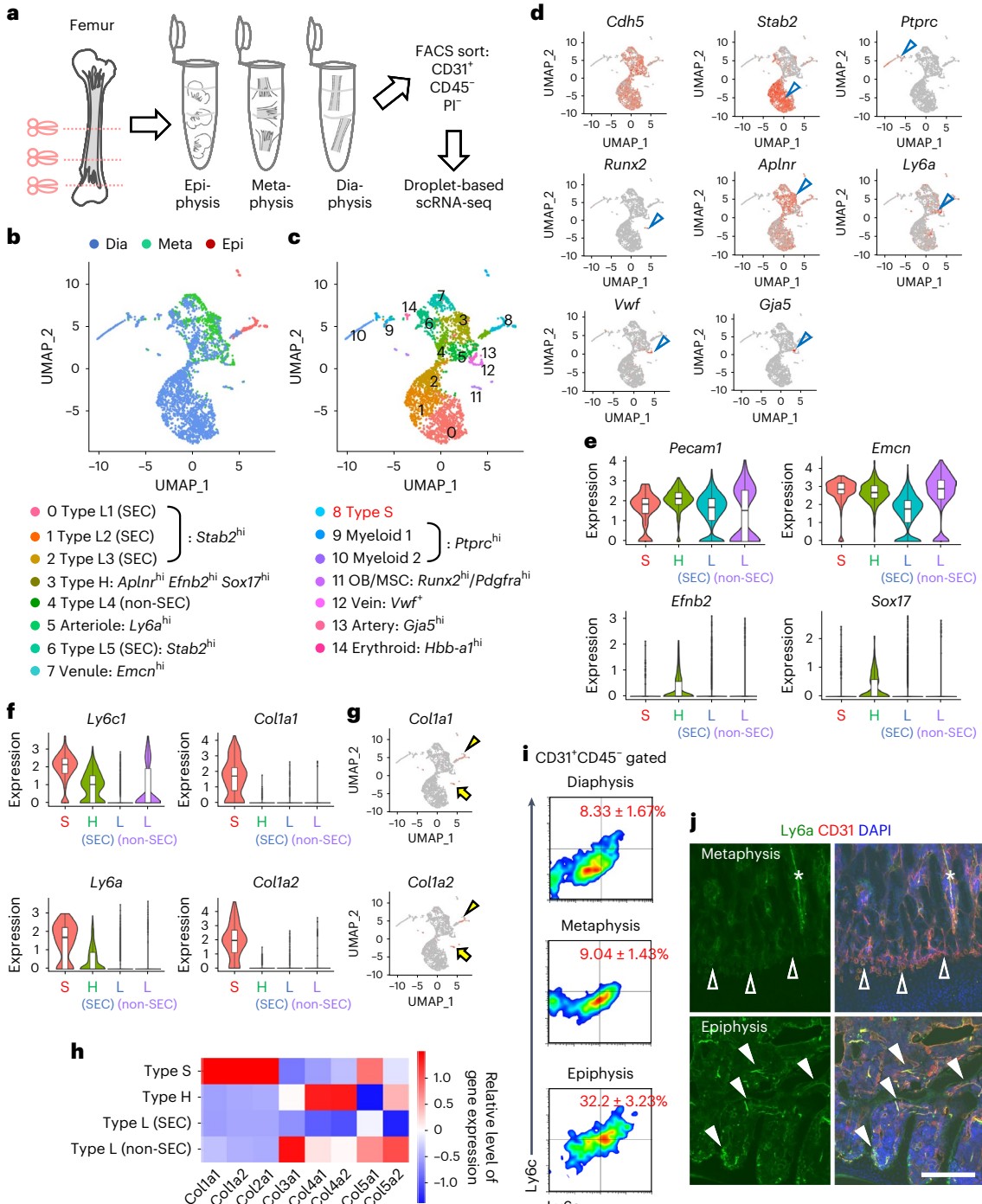

**Fig. 1 | Spatial heterogeneity of transcriptomes in BMECs. a**, Schematic diagram depicting the strategy for scRNA-seq of BMECs. **b–d**, Feature plots of BMECs of three bone fragments and clustering based on the expression of canonical endothelial subtype markers. Dia, diaphysis; Epi, epiphysis; Meta, metaphysis. **b**, Feature plot showing all cells derived from three datasets (epiphysis, metaphysis and diaphysis). **c**, Feature plot showing all cells derived from three datasets (epiphysis, metaphysis and diaphysis) with annotation for each cluster. **d**, Feature plots showing expression of marker genes with enriched expression for each endothelial cluster. The arrowheads in **d** indicate clusters represented by each marker. **e**, Violin plots showing the expression of pan-endothelial and type H markers. **f**, Violin plots showing the expression of genes enriched in type S ECs. **g**, Feature plots of BMECs. *Col1a1* and *Col1a2* genes are highly expressed in type S ECs (arrowheads), although their expression is highest in OBs/MSCs (arrows). **h**, A heatmap showing expression of various

collagen genes. **i**, Flow cytometric analysis of isolated CD31⁺CD45⁻ BM cells at P11 (Ly6a^high^Ly6c^high^ surface protein levels) (*n* = 4, biologically independent experiments). The *P* value for the epiphysis versus the metaphysis is 0.000012. **j**, Immunohistochemistry of femur sections at P11. Ly6a is abundantly expressed in type S ECs (closed arrowheads) and AECs (asterisks), but weakly in type H ECs (open arrowheads). In the box plots of **e** and **f** the lower and upper bounds of box represent the 25th and 75th percentiles, respectively; the box represents the middle 50% of the data, namely the interquartile range (IQR); the horizontal line within the box represents the median; and the lower and upper bounds of the whiskers represent the minimum and maximum values within 1.5× the IQR below the 25th percentile and above the 75th percentile, respectively. The black dots represent outliers. Scale bar, 200 μm. Data presented are the mean ± s.d. The comparisons between the averages of two groups were evaluated using the two-tailed Student's *t*-test.

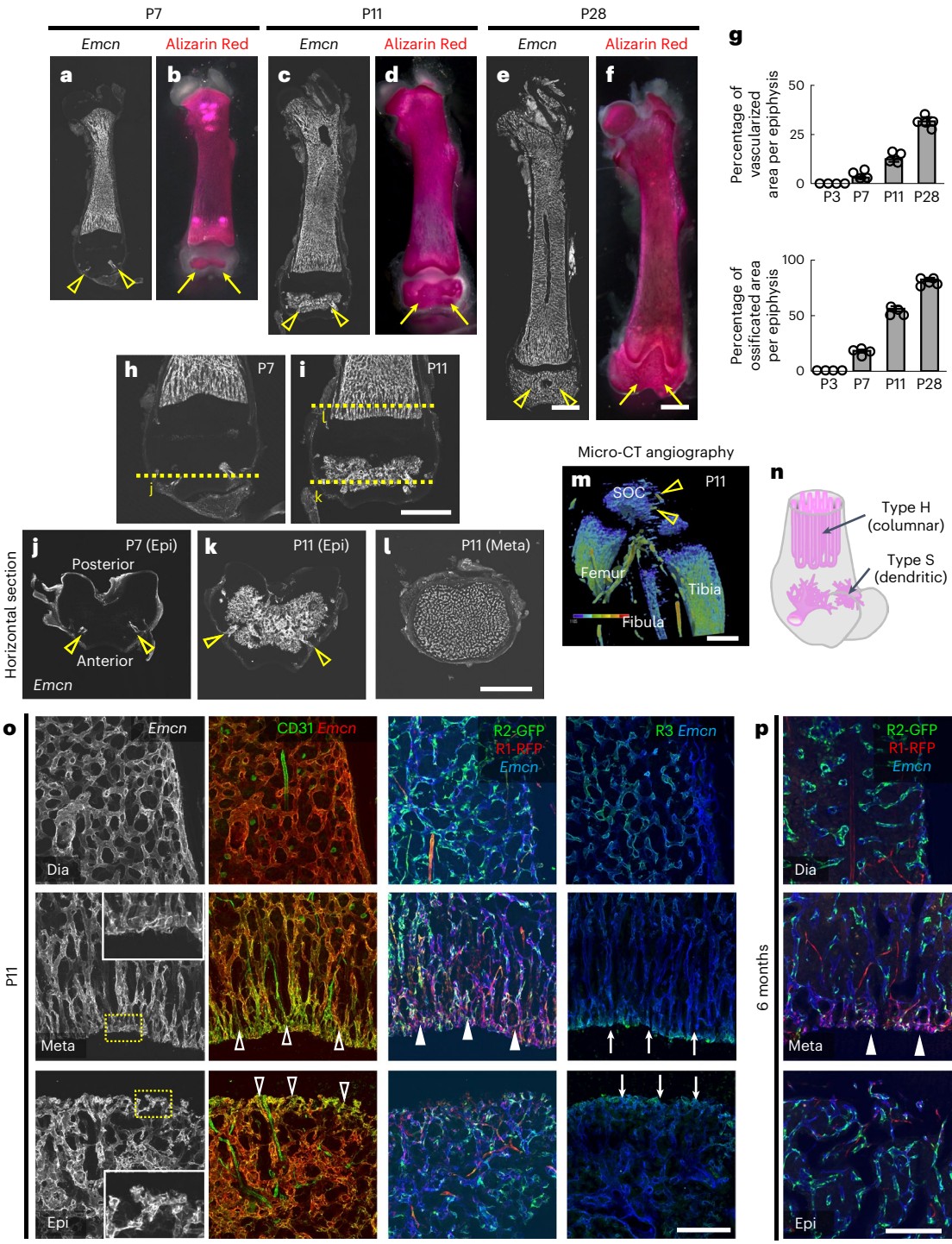

**Fig. 2 | Molecular and morphological features of type S ECs. a–f**, Whole-mount Alizarin Red staining (**b** at P7, (**d**) at P11 and **f** at P28) and section immunostaining of femurs (**a** at P7, **c** at P11 and **e** at P28). Vessel invasion of the epiphysis (open arrowheads) and SOC formation (arrows) occur at P7 and spread three-dimensionally during the second week after birth. **g**, Quantification of vessels and calcified areas in the epiphysis (P3, P7 and P11: *n* = 4; P28: *n* = 5, biologically independent experiments). **h–l**, Immunohistochemistry of femurs (vertical (**h** at P7 and **i** at P11) or horizontal sections **j** at P7 (epiphysis), **k** at P11 (epiphysis) and **l** at P11 (metaphysis)). Open arrowheads indicate vessel invasion points. **m**, Micro-CT angiography at P11 showing the heatmap of vessel perfusion of the knee joint. Open arrowheads indicate vessel invasion points. **n**, Schematic diagram showing the structural difference between type H (columnar) and type S (dendritic) vessels. **o**,**p**, Immunohistochemistry of femur sections at P11 (**o**) or 6 months after birth (**p**). The growing edges of type S and type H vessels highly express CD31 (open arrowheads) and Vegfr3 (arrows), and project short filopodia towards avascular areas (insets in the leftmost panels). Vegfr1 is specifically expressed in type H ECs (closed arrowheads). R3, Vegfr3; R2-GFP, GFP in Vegfr2-GFP transgenic mice; R1-RFP, red fluorescent protein (RFP) in Vegfr1-RFP transgenic mice. Scale bars, 1 mm (**a–f**, **h–l** and **m**); 200 μm (**o** and **p**). Data presented are the mean ± s.d.

completely depleted[30], did not show apparent vascular defects in the epiphysis, suggesting that osteoclasts are not relevant to vascularization of this area (Extended Data Fig. 4). Taken together, these data show that type S vessels are highly plastic during postnatal development, even more plastic than type H vessels, and robustly contribute to epiphyseal osteogenesis.

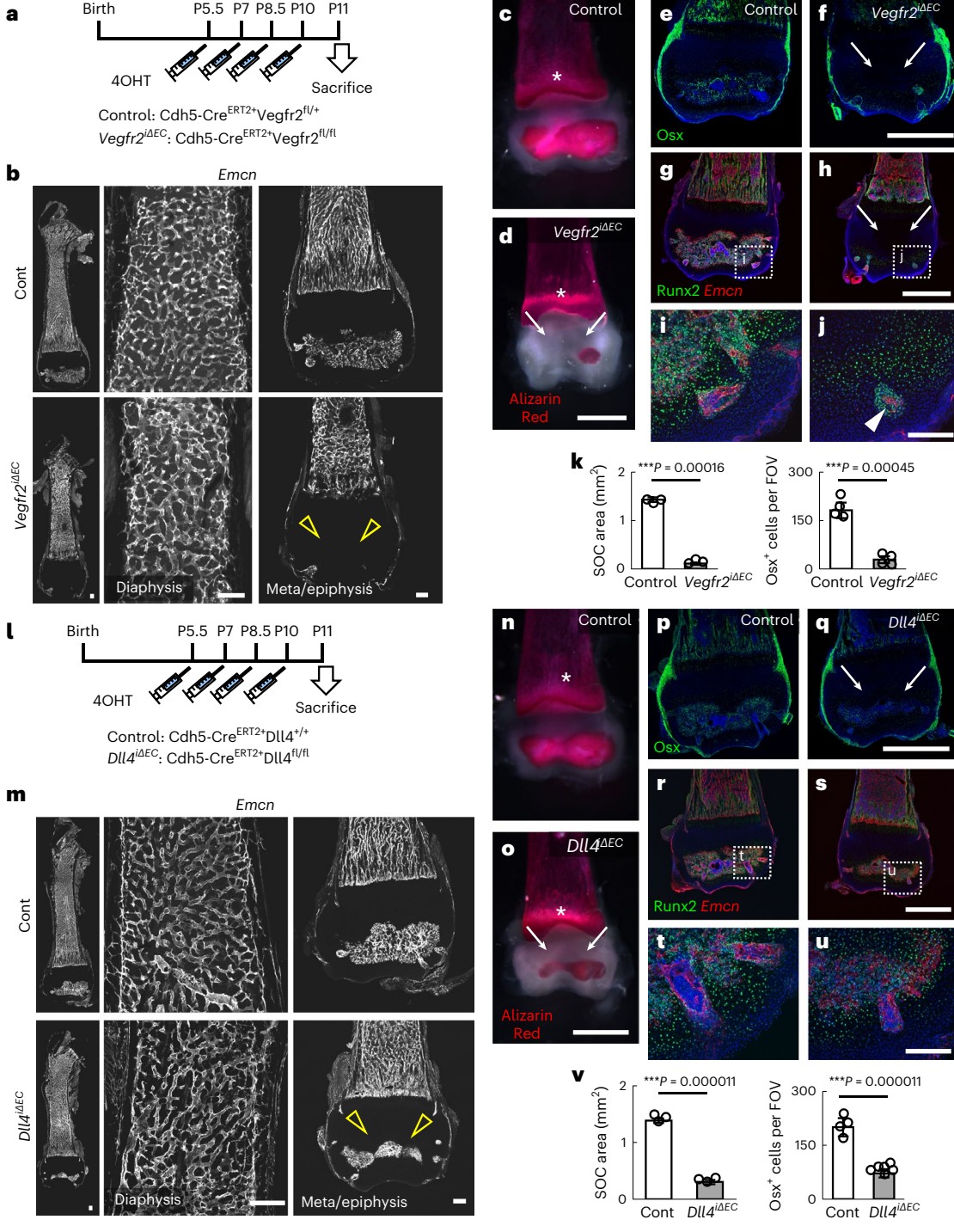

**Fig. 3 | Type S vessels are highly plastic and crucial for osteogenesis.**
**a**, Protocol for 4OHT injection in neonates. **b–d**, Femur sections (**b**) and whole-mount Alizarin Red staining (control (**c**) and *Vegfr2*$^{i\Delta EC}$ (**d**)) at P11. *Vegfr2*$^{i\Delta EC}$ mice show severely impaired angiogenesis (open arrowheads) and osteogenesis (arrows) in the epiphysis, although metaphyseal ossification is largely intact (asterisks). **e–j**, Immunohistochemistry of femur sections stained with Osx and DAPI (**e,f**) or Runx2 and Emcn (**g–j**). Panels **i** and **j** how the high magnification views of the area indicated in **g** and **h**. Panels **e**, **g** and **i** are from control, and panels **f**, **h** and **j** are from *Vegfr2*$^{i\Delta EC}$ mice. Osx$^+$ and Runx2$^+$ cells are found around vessels, their numbers are reduced (arrows) and they are detected only around surviving vessels (arrowhead) in *Vegfr2*$^{i\Delta EC}$ mice. **k**, Quantification of the SOC area (*n* = 3, biologically independent experiments) and Osx$^+$ cells (*n* = 4, biologically independent experiments) in the epiphysis. **l**, Protocol for 4OHT injection in neonates. **m–o**, Femur sections (**m**) and whole-mount Alizarin Red staining

(control (**n**) and *Dll4*$^{i\Delta EC}$ (**o**)) at P11. *Dll4*$^{i\Delta EC}$ mice have an increased vessel density but a reduced vascularized area (open arrowheads) and reduced osteogenesis (arrows) in the epiphysis, although metaphyseal ossification is largely intact (asterisks). **p–u**, Immunohistochemistry of femur sections stained with Osx and DAPI (**p,q**) or Runx2 and Emcn (**r–u**). Panels **t** and **u** show the high magnification views of the area indicated in **r** and **s**. Panels **p**, **r** and **t** are from control, and panels **q**, **s** and **u** are from *Dll4*$^{i\Delta EC}$ mice. Osx$^+$ and Runx2$^+$ cells are found around vessels and their numbers are reduced (arrows) in *Dll4*$^{i\Delta EC}$ mice. **v**, Quantification of the SOC area (*n* = 3, biologically independent experiments) and Osx$^+$ cells (control: *n* = 4, *Dll4*$^{i\Delta EC}$: *n* = 6, biologically independent experiments) in the epiphysis. Data are presented as the mean ± s.d. ***$P < 0.001$; two-tailed Student's *t*-test. Scale bars, 1 mm (**c–h** and **n–s**); 200 μm (**b,i,j,m,t,u**). The comparisons between the averages of two groups were evaluated using the two-tailed Student's *t*-test.

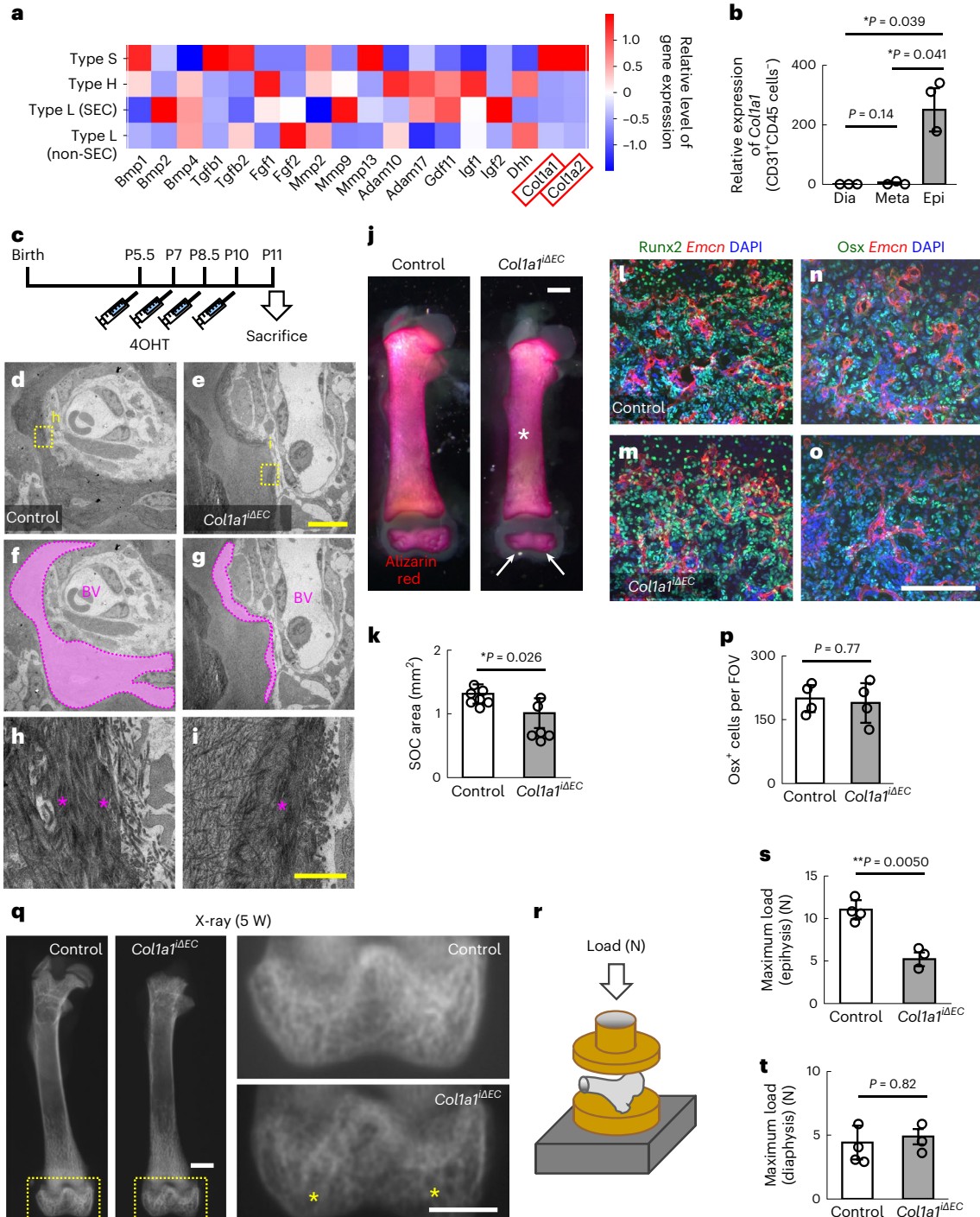

**Fig. 4 | Type S vessels contribute to osteogenesis by secreting type I collagen.**
**a**, A heatmap showing osteogenic factors based on the scRNA-seq data shown in Fig. 1. **b**, Quantitative PCR analysis of CD31$^+$CD45$^-$ cells derived from femurs of P11 mice ($n = 3$, biologically independent experiments). **c**, Protocol for 4OHT injection in neonates. **d**–**i**, TEM of epiphyseal sections of P11 mice. Panels **d**, **f** and **h** are from control, and **e**, **g** and **i** are from *Col1a1*$^{iΔEC}$ mice. Panels **f** and **g** show the area of perivascular collagen fibres with dotted area of **d** and **e**. Panels **h** and **i** show high magnification views of the area indicated in **d** and **e**, respectively. The bands of collagen fibres around blood vessels (dotted area) are thin and sparse in *Col1a1*$^{iΔEC}$ mice (asterisks). **j**,**k**, Whole-mount Alizarin Red staining of femurs at P11 (**j**) and quantification (**k**) (control: $n = 7$; *Col1a1*$^{iΔEC}$: $n = 6$, biologically independent experiments). Arrows indicate reduced mineralization in the epiphyses and the asterisk indicates intact ossification in the diaphyses and metaphyses of

*Col1a1*$^{iΔEC}$ mice. **l**–**p**, Immunohistochemistry of femur sections at P11 stained with Runx2, Emcn and DAPI (**l**,**m**) or Osx, Emcn and DAPI (**n**,**o**). Panels **l** and **n** are from control, and **m** and **o** are from *Col1a1*$^{iΔEC}$ mice. Quantification of Osx$^+$ cells ($n = 4$, biologically independent experiments) is shown in **p**. **q**, X-ray images of adult mice. *Col1a1*$^{iΔEC}$ mice show reduced X-ray intensity (asterisks) in the epiphysis. **r**–**t**, Mechanical strength of the femoral distal epiphysis or diaphysis measured by a compression test. Panel **r** shows the scheme of the test, and quantification of the maximum load of the epiphysis (**s**) and the metaphysis (**t**) is shown individually (control: $n = 4$; *Col1a1*$^{iΔEC}$: $n = 3$, biologically independent experiments). The epiphysis has inferior biomechanical properties in *Col1a1*$^{iΔEC}$ mice. Scale bars, 1 mm (**j** and **q**); 200 µm (**l**–**o**); 10 µm (**d**–**g**); and 1 µm (**h** and **i**). $^*P < 0.05$; data presented are the mean ± s.d. The comparisons between the averages of two groups were evaluated using the two-tailed Student's $t$-test.

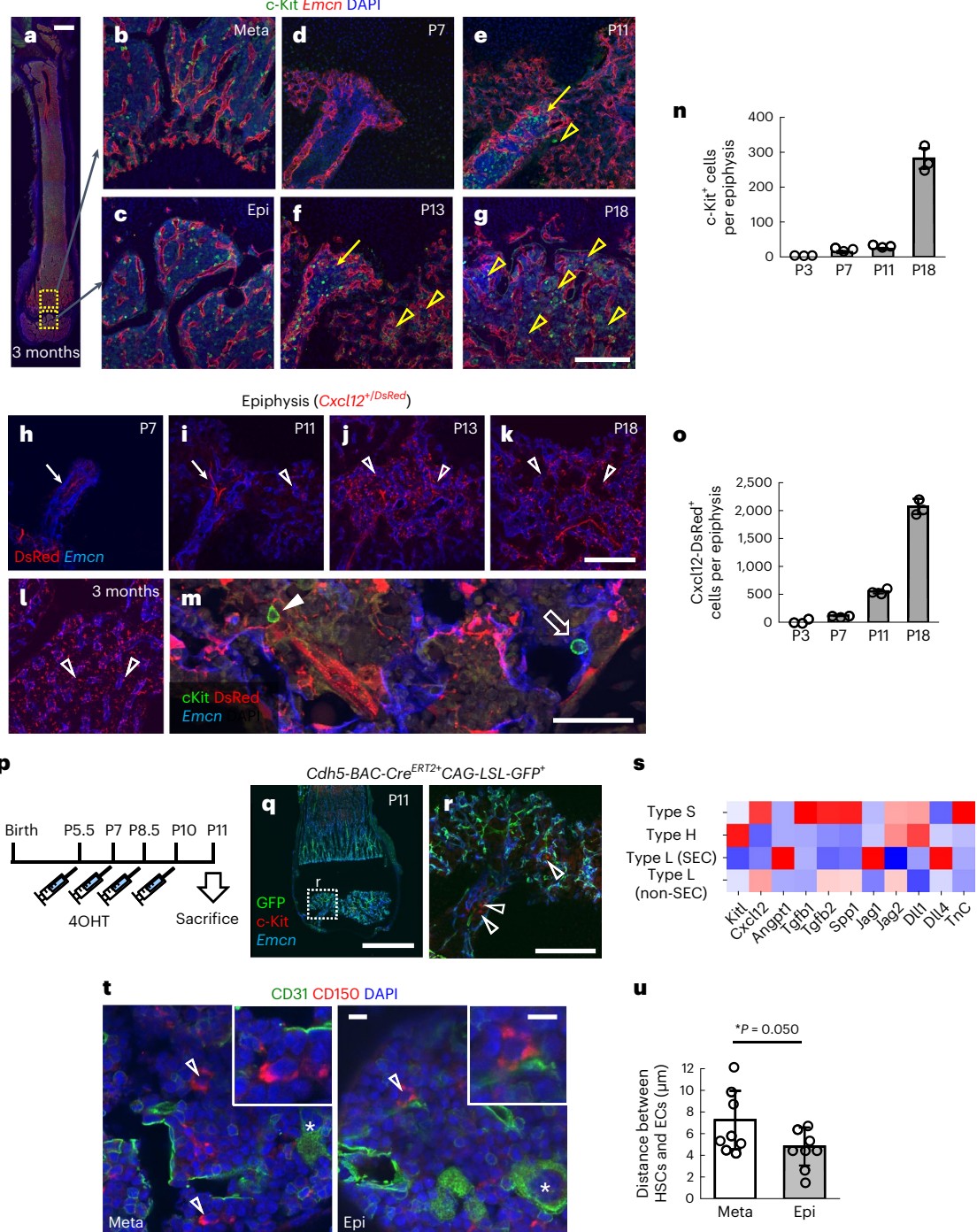

**Fig. 5 | Type S vessels postnatally establish a satellite niche for HSPCs.**
**a–g**, Immunohistochemistry of femur sections at 3 months after birth (**a–c**), P7 (**d**), P11 (**e**), P13 (**f**) and P18 (**g**). Panels **b** and **c** show the high magnification views of the area indicated in **a**. The c-Kit⁺ cells start to appear in the entry point of type S vessels around P11 (arrows) and gradually spread in the epiphysis thereafter (arrowheads). **h–m**, Immunohistochemistry of femur sections from *Cxcl12⁺/DsRed* mice at P7 (**h**), P11 (**i**), P13 (**j**) and P18 (**k**), and 3 months after birth (**l,m**). Panel **m** shows the high magnification view co-stained with cKit and Emcn. The arrows indicate DsRed expression detected in arteries. MSCs appear at P11 and spread throughout the epiphysis thereafter (open arrowheads). Most c-Kit⁺ cells are located in contact with MSCs (closed arrowhead) or ECs (open arrow). **n,o**, Quantification of c-Kit⁺ cells (**n**) and Cxcl12-DsRed⁺ cells (**o**) in BM sections at various stages (*n* = 3, biologically independent experiments). **p**, Protocol for

4OHT injection in neonates. **q,r**, Immunohistochemistry of femur sections. Panel **r** shows the high magnification view of the area indicated in **q**. Lineage tracing using *Cdh5-BAC-Cre^ERT2* mice shows that c-Kit⁺ cells (arrowheads) do not originate from haemogenic endothelium in type S vessels. **s**, A heatmap showing HSPC niche factors based on the scRNA-seq data shown in Fig. 1. **t,u**, Immunohistochemistry of femur sections from a mouse at 3 months (**t**) and quantification (**u**) (*n* = 8, biologically independent experiments). The distance between HSCs (open arrowheads) and ECs is smaller in epiphyses than in metaphyses. Asterisks are CD31⁺ megakaryocytes. Scale bars, 1 mm (**a** and **q**); 200 μm (**b–g** and **h–l,r**); 50 μm (**m**); and 10 μm (**t**). Data presented are the mean ± s.d. *P < 0.05. The comparisons between the averages of two groups were evaluated using the two-tailed Student's *t*-test.

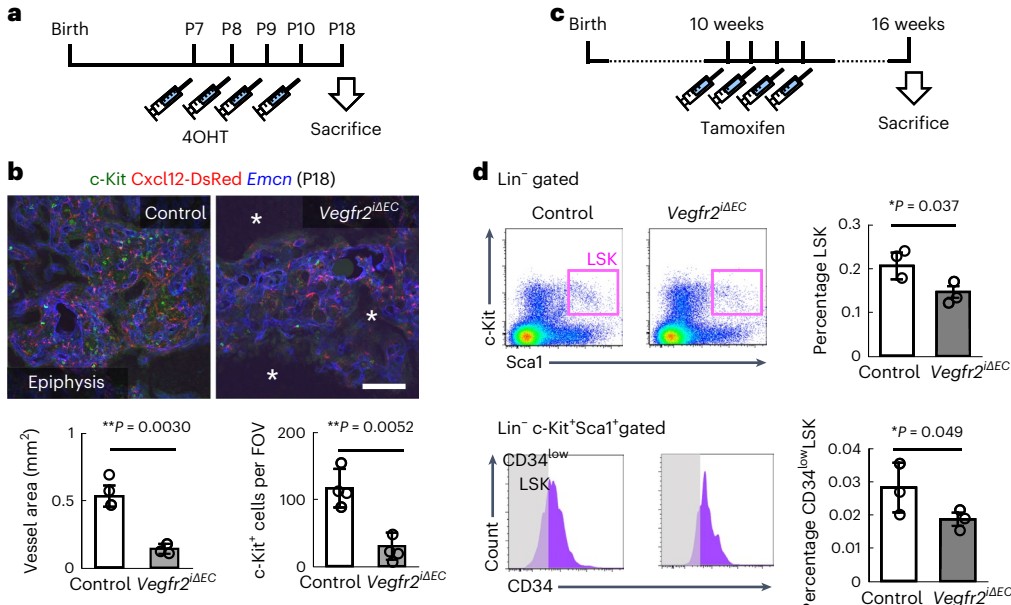

**Fig. 6 | Type S vessels harbour HSCs. a**, Protocol for 4OHT injection during the second week after birth. **b**, Immunohistochemistry of femur sections at P18 and quantification (vessel area: $n = 3$, c-Kit$^+$ cells per FOV: $n = 4$, biologically independent experiments). *Vegfr2$^{i\Delta EC}$* mice lack c-Kit$^+$ cells especially in the avascular area (asterisks). **c**, Protocol for tamoxifen injection in adult mice.

**d**, Flow cytometric analysis of BM cells isolated from the epiphysis of 12-week-old mice and quantification ($n = 3$, biologically independent experiments). Scale bar, 200 μm. $^{**}P < 0.01$; $^*P < 0.05$. Data presented are the mean ± s.d. The comparisons between the averages of two groups were evaluated using the two-tailed Student's *t*-test.

## Type S vessels regulate osteogenesis by secreting collagen I

Considering the data described above, we were interested in how type S vessels contribute to epiphyseal osteogenesis. Type S ECs did not apparently preferentially express canonical osteogenic factors such as bone morphogenetic proteins (Bmps), fibroblast growth factors (Fgfs) and insulin-like growth factors (Igfs), in contrast with their specific expression of *Col1a1* and *Col1a2* (Fig. 4a). This expression pattern of *Col1a1* and *Col1a2* was confirmed by quantitative (q)PCR analysis of sorted BMECs (Fig. 4b). Bone is a hybrid tissue with two main components: collagen fibril frameworks and calcium phosphate minerals. Type I collagen is the most abundant protein constituting collagen fibrils in bones. Therefore, we generated endothelial-specific *Col1a1*-knockout (*Cdh5-BAC-Cre$^{ERT2+}$Col1a1$^{flox/flox}$*, hereafter referred to as *Col1a1$^{i\Delta EC}$*) mice and analysed the collagen pattern in epiphyseal bone tissues by transmission electron microscopy (TEM) at P11 (Fig. 4c). As expected, the bands of collagen fibres around blood vessels were thin and sparse in *Col1a1$^{i\Delta EC}$* mice (Fig. 4d–i). It is interesting that mineral deposition evaluated by Alizarin Red staining was significantly reduced in the epiphysis, but not in other bone portions, of *Col1a1$^{i\Delta EC}$* mice (Fig. 4j,k). The numbers of pre-osteoprogenitors and osteoprogenitors were not significantly affected in *Col1a1$^{i\Delta EC}$* mice (Fig. 4l–p); therefore, insufficient collagen frameworks might impair mineral deposition or stability[31]. At 5 weeks after birth, *Col1a1$^{i\Delta EC}$* mice showed reduced X-ray intensity (Fig. 4q), reflecting impaired mineral deposition. Accordingly, bone strength evaluated by the compression test was significantly reduced in the epiphysis, but not in the diaphysis, of *Col1a1$^{i\Delta EC}$* mice (Fig. 4r–t). These data suggest that mesenchymal cell-derived type I collagen is insufficient for epiphyseal osteogenesis, but ECs complement this insufficiency by secreting type I collagen.

## Type S vessels establish a satellite niche for HSPCs

In haematopoietic analysis of BM cells, epiphyses are usually discarded during the process of flushing out the BM cavity. Therefore, the emergence and residence of HSPCs in epiphyses have not been intensively studied. Although Sca1 staining only marked endothelial cells (Fig. 1j),

visualization of putative HSPCs by c-Kit immunostaining (Fig. 5a–g,n) showed that these cells started to appear in the entry point of type S vessels around P11 (Fig. 5d,e). Thereafter, they expanded throughout the BM cavity in the epiphysis until weaning (Fig. 5c,f,g). Next, we investigated MSCs, one of the most important components of the HSPC niche[11,19], using *Cxcl12$^{+/DsRed}$* mice. Although DsRed expression in these mice was detected in only a few AECs before P7, it was robustly detected in perivascular reticular cells after P13, suggesting that the emergence of MSCs synchronizes with the growth of type S vessels (Fig. 5h–o and Extended Data Fig. 5a–g). HSPCs were located in close proximity to type S ECs or MSCs (Fig. 5m), suggesting the existence of vascular and mesenchymal HSPC niches[11]. Lineage tracing using *Cdh5-BAC-Cre$^{ERT2}$* mice showed that c-Kit$^+$ HSPCs were not labelled, excluding the possibility of their de novo specification from haemogenic endothelium (Fig. 5p–r). Rather, they were supposedly derived from circulating HSPCs in peripheral blood. Reanalysis of scRNA-seq data found that type S ECs showed moderate but higher expression of some HSPC niche factors such as *Cxcl12*, *Tgfb1*, *Tgfb2*, *Spp1* (osteopontin) and *TnC* than type H ECs (Fig. 5s). We also visualized haematopoietic stem cells (HSCs) by performing CD150 staining via modification of a protocol reported by other groups[25,32,33], and found that the distance between HSCs and ECs was smaller in epiphyses than in metaphyses (Fig. 5t,u). Consistently, the number of c-Kit$^+$ HSPCs in epiphyses was significantly reduced in BM sections of *Vegfr2$^{i\Delta EC}$* mice at P18 (Fig. 6a,b). It is of interest that MSCs were associated with type S ECs and thus the number of these cells was also decreased in *Vegfr2$^{i\Delta EC}$* mice (Fig. 6b). Flow cytometric analysis showed that the numbers of lineage$^-$Sca1$^+$c-Kit$^+$ (LSK) HSPCs and CD34$^{low}$LSK HSCs were significantly decreased in adult *Vegfr2$^{i\Delta EC}$* mice (Fig. 6c,d). Taken together, these data demonstrate that type S vessels postnatally establish a satellite niche for HSPCs that is distant from the main BM cavity.

## Epiphyseal HSCs reconstitute long-term haematopoiesis

Stromal cells in the metaphysis, including osteogenic cells and type H ECs, provide a favourable microenvironment for dormant HSCs[14,24,34].

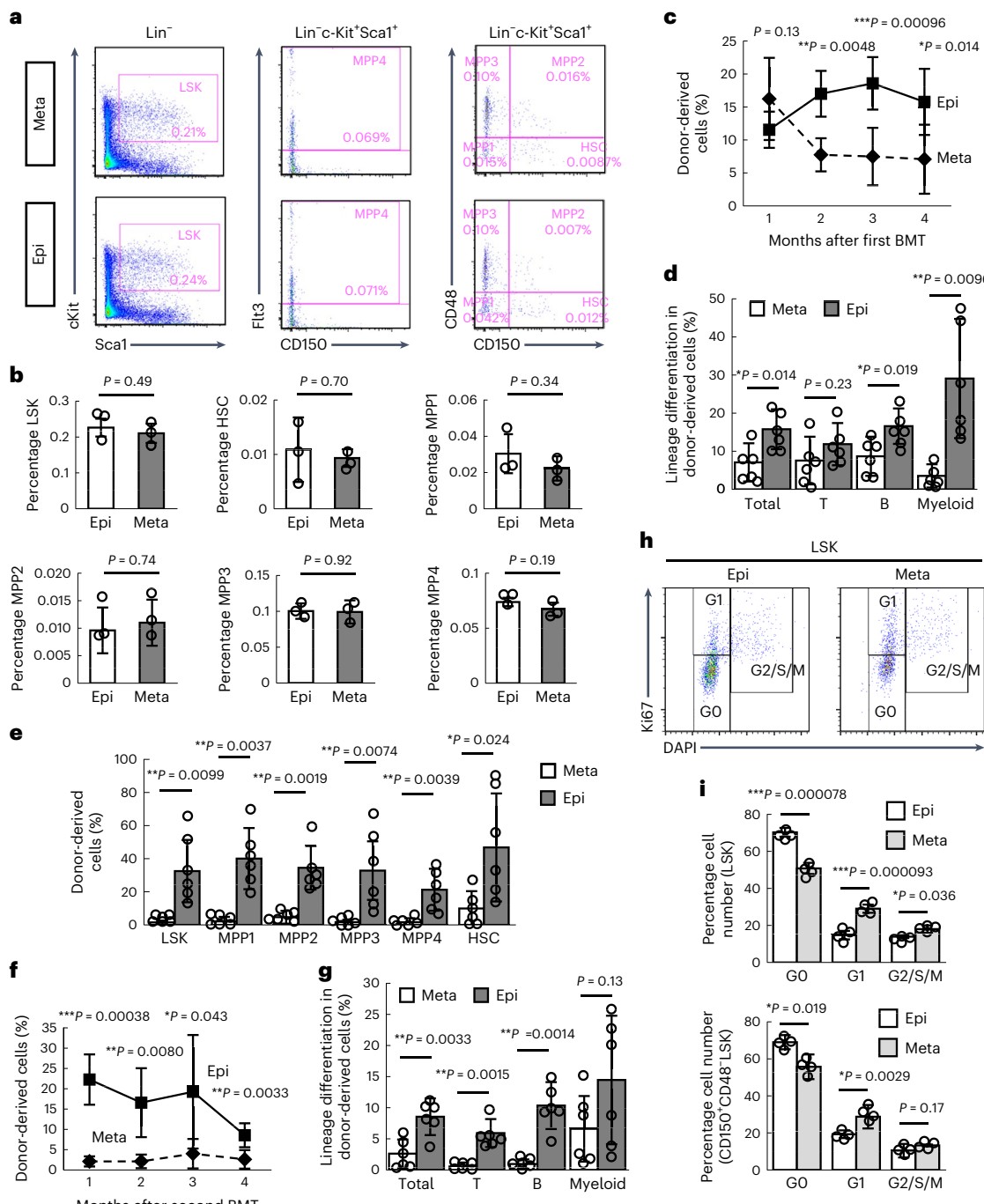

**Fig. 7 | Epiphysial HSCs reconstitute long-term multilineage haematopoiesis.**
**a,b**, Flow cytometric analysis of BM cells isolated from the epiphysis and metaphysis of 12-week-old mice (**a**) and quantification (**b**) ($n = 3$, biologically independent experiments). **c,d**, Quantification of donor-derived cells (**c**) and lineage differentiation in peripheral blood samples (**d**) after the first BMT ($n = 6$, biologically independent experiments). **e**, Chimaerism of donor-derived cells in the BM ($n = 6$, biologically independent experiments) at 4 months after the first BMT. **f,g**, Quantification of donor-derived cells (**f**) and lineage differentiation in peripheral blood samples (**g**) after the second BMT ($n = 6$, biologically independent experiments). **h,i**, Flow cytometric analysis of BM cells isolated from the epiphysis and metaphysis of 12-week-old mice (**h**) and quantification (**i**) ($n = 4$, biologically independent experiments). The cell-cycle state was determined by DAPI and Ki67 staining. $^{***}P < 0.001$; $^{**}P < 0.01$; $^{*}P < 0.05$. Data presented are the mean ± s.d. The comparisons between the averages of two groups were evaluated using the two-tailed Student's $t$-test.

This prompted us to compare the ability of metaphyses and epiphyses to function as an HSPC niche. Flow cytometric analysis showed that there were comparable numbers of HSPCs and HSCs in metaphyses and epiphyses (Fig. 7a,b). Next, we conducted a bone marrow transplantation (BMT) and compared the haematopoietic reconstitution activity of epiphyseal and metaphyseal HSCs. First, donor cells ($4 \times 10^5$ BM mononuclear cells (BMMNCs)) isolated from the epiphysis or metaphysis of Ly5.2 mice were transplanted into lethally irradiated Ly5.1 mice ($n = 6$ per group). Although there was no difference in chimaerism in peripheral blood between metaphyseal and epiphyseal cells during the first month, epiphyseal cells showed higher reconstitution activity than metaphyseal cells at 2 months after transplantation (Fig. 7c,d).

The frequency of donor-derived HSPCs in the BM was also higher in mice transplanted with epiphyseal cells than in mice transplanted with metaphyseal cells (Fig. 7e). This difference in reconstitution activity between epiphyseal and metaphyseal cells became much more evident when BMMNCs were serially transplanted into secondary recipient mice (Fig. 7f,g). Next, we evaluated the cell-cycle state in HSPCs and HSCs (Fig. 7h,i). More HSPCs and HSCs were in G0 phase in epiphyses than in metaphyses, suggesting that epiphyseal HSCs are relatively dormant, which probably accounts for their superior BM reconstitution ability without a difference in phenotypic characteristics in flow cytometric analysis. Moreover, we assessed the functional capacities of single HSCs by performing limiting dilution analysis of donor HSCs from metaphyses and epiphyses. At 3 months after transplantation, the engraftment rate of donor cells from epiphyses was 2.5× higher than that of donor cells from metaphyses, although the difference was not significant ($P = 0.19$; $n = 8$ recipients per group) (Extended Data Fig. 6). These data suggest that type S vessels constitute a distinct and efficient reservoir for HSCs separate from the main BM cavity.

## Discussion

In the present study, using scRNA-seq, we thoroughly characterized a spatial heterogeneity of transcriptomes in BMECs. During the analysis, we identified a capillary subtype found exclusively in the epiphysis. These cells possessed unique structural and gene expression features and robustly contributed to bone strength as well as haematopoietic maintenance.

Previous studies analysing BM cells by scRNA-seq uncovered the cellular taxonomy of stromal compartments such as mesenchymal stem/stromal cells, chondrocytes, fibroblasts, BMECs and pericytes[15,16]. In these studies, BMECs were largely divided into two populations: AECs and SECs. Our current analysis annotated 11 EC subclusters. These previous studies did not recognize some EC populations, in particular type H ECs, which are probably included in the AEC cluster because they used Ly6a expression as a hallmark of AECs. It should be noted that the BM was flushed and enzymatically digested after the epiphyses were cut out and therefore type S ECs were inevitably discarded in these studies.

Recent discoveries in the field of bone metabolism greatly improved our understanding of angiogenic–osteogenic coupling, especially in endochondral ossification of long bones[6–9,22,23]. Type H vessels promote bone formation and contribute to the maintenance of HSPCs[14,23]. Pharmacological augmentation of type H vessels enhances bone fracture healing and counteracts bone loss in ovariectomy-induced osteoporotic mice[35]. Our current study also underscores the importance of type S vessels, especially in the ossification of epiphyses. In terms of physical activity, type S ECs are critically important because epiphyses sustain the stability of joints, which often becomes problematic in ageing and various bone diseases such as osteoporosis and osteonecrosis. On the other hand, from the viewpoint of systemic haematopoietic maintenance, the contribution of type S ECs might be minimal considering the small size of epiphyseal BM. Although our experiments showed that type S ECs harbour transplantable HSCs, epiphyses probably act as a satellite niche, which might be utilized in an emergency such as extensive loss of the main BM cavity. In general, BMECs have two functional aspects: haematopoiesis and osteogenesis. Type S ECs may prioritize the latter compared with BMECs in other fragments, although it is unclear which biological factor regulates their prioritization of osteogenesis.

Type I collagen is produced by mesenchymal cells such as OBs and fibroblasts. Our data showing that ECs are an important source of type I collagen specifically in epiphyses raise the following question: why do epiphyses employ such a unique ossification system? We speculate that the mode of epiphyseal growth answers this question. Epiphyses undergo rapid ossification and angiogenesis synchronously in the postnatal stage, when the joints are already used daily and therefore epiphyses are exposed to friction forces. In this complicated and emergent process, it is speculated that type S ECs participate in matrix deposition, which is conducted by OBs alone in other fragments. Such a function of ECs could be a unique angiocrine mode[3], that is, constructing the framework required for organ functions. However, it should be mentioned that the phenotype of *Vegfr2$^{i\Delta EC}$* mice was much more severe than that of *Col1a1$^{i\Delta EC}$* mice, suggesting that the contribution of type S ECs to osteogenesis cannot be solely attributed to type I collagen secretion. Other angiocrine factors regulating osteogenesis[36] or delivery of oxygen and nutrients might account for the phenotypic gap between these two mutant lines.

It is unclear how type S ECs preferentially produce type I collagen compared with other collagens and what makes type S ECs distinct from other EC subtypes. Epigenetic regulation might account for this unique expression in type S ECs. Moreover, type S ECs might have some characteristic features in multiple steps of collagen synthesis such as posttranslational modification in the endoplasmic reticulum, release from the ribosome, glycosylation of procollagens and secretion[37].

In the future, it is important to determine whether type S ECs are present in humans because epiphyseal blood vessels are critically involved in human diseases such as Legg–Calvé–Perthes disease (LCPD), a type of femoral head necrosis that affects children. There is no known cause of impaired blood supply to the femoral head in LCPD[38].

Overall, our current study provides new insight into the landscape for the cellular architecture and heterogeneity of BMECs and uncovers the importance of epiphyseal ECs, which has been overlooked so far. In the future, our findings may lead to the discovery of new therapeutic options for bone diseases such as LCPD and osteoporosis.

## Online content

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

## Methods

### Mice

Animal experiments were approved by the Institutional Animal Care and Use Committee of Keio University (protocol no. A2021-003) and were performed in accordance with the Guidelines of Keio University for Animal and Recombinant DNA Experiments (protocol no. D2005-044). *Cdh5-BAC-Cre^ERT2* (ref. [39]), *Osx-Cre^ERT2* (ref. [8]), *Vegfr2-flox*[40], *Dll4-flox*[41], *Col1a1-flox*[42], *CAG-LSL-EGFP*[43], *Vegfr1-BAC-DsRed*[44], *Vegfr2-BAC-GFP*[44] and *Rbpj-flox*[45] mice had been developed previously. All mice were crossed with C57BL/6J mice more than 8× and maintained. For analyses of phenotypes, mice of both sexes were used. *Csf1^op/op* (stock no. 000231) and *Cxcl12^+/DsRed* (stock no. 0225458) mice were obtained from the Jackson Laboratory. For neonatal experiments of tamoxifen-inducible expression of Cre, 40 μg of 4-hydroxytamoxifen (4OHT, Sigma-Aldrich, catalogue no. H6278) was subcutaneously injected at the indicated timepoints. For adult experiments of tamoxifen-inducible expression of Cre, 1 mg of tamoxifen (Sigma-Aldrich, catalogue no. H5648) was subcutaneously injected at the indicated timepoints.

### Preparation of tissue sections

Samples were collected and placed in 4% paraformaldehyde at 4 °C for 16–24 h under rotation. Decalcification was performed with 0.5 M EDTA for 16–24 h at 4 °C under rotation. Decalcified bone was infiltrated with 30% sucrose for 2 d under rotation, embedded with optimal cutting temperature compound and sectioned at a thickness of 100 μm using tungsten carbide knives (Thermo Fisher Scientific).

### Immunostaining

Sections were air-dried at room temperature, rehydrated in phosphate-buffered saline (PBS) and then blocked using 2% bovine serum albumin (BSA) overnight at 4 °C. Sections were incubated with primary antibodies overnight at 4 °C in a humidified chamber and then with secondary antibodies overnight at 4 °C in a humidified chamber. After thorough washing, sections were mounted. The following primary monoclonal antibodies were used: anti-CD31 (Abcam, catalogue no. ab119341; 1:1,000), anti-Runx2 (Abcam, catalogue no. ab192256, 1:500), anti-ASMA (Sigma-Aldrich, catalogue no. A5228; 1:200), anti-Ly6a (BD Biosciences, catalogue no. 553333; 1:500), anti-endomucin (Santa Cruz, catalogue no. sc-65495; 1:500) and anti-CD150 (BioLegend, catalogue no. 115905; 1:200). The following primary polyclonal antibodies were used: anti-green fluorescent protein (GFP)-Alexa Fluor-488 (Molecular Probes, catalogue no. A21311, 1:500), anti-c-Kit (R&D Systems, catalogue no. AF1356, 1:500), anti-Sp7/Osx (Abcam, catalogue no. ab22552, 1:500), anti-cathepsin K (Abcam, catalogue no. ab19027, 1:500), anti-VEGFR3 (R&D Systems, catalogue no. AF743, 1:1000), anti-aggrecan (Sigma-Aldrich, catalogue no. AB1031, 1:500), anti-CGRP (Sigma-Aldrich, catalogue no. C8198, 1:8000) and anti-Erg (Abcam, catalogue no. ab92513, 1:2,000). The secondary antibodies used were Alexa Fluor-488-conjugated immunoglobulin (Ig)G (Molecular Probes, catalogue nos. A11034, A11006, and A11055; 1:500) and Cy3/Cy5 DyLight549/DyLight649-conjugated IgG (Jackson ImmunoResearch, catalogue nos. 711-165-152, 112-165-167, 127-165-160, 711-605-152, 112-605-167 and 127-605-160; 1:500). For nuclear staining, specimens were treated with DAPI (Molecular Probes, catalogue no. D-1306). For immunostaining of both intracellular and extracellular aggrecan, slides were pretreated with hyaluronidase (Sigma-Aldrich, catalogue no. H4272, 2 mg ml⁻¹) at 37 °C for 30 min before primary antibody staining.

### Confocal microscopy

Fluorescent images were obtained using a confocal laser scanning microscope (Olympus, FV3000). Obtained images were analysed using FV10-ASW 3.0 Viewer (Olympus). Cells or substances of interest were quantified in the entire area of the epiphysis, in a 600 × 600 μm² field of view (FOV; to count OSX⁺ and c-Kit⁺ cells and to quantify relative intensity of immunofluorescence), or in a 1-mm length of the vascular front (to count tip-cell filopodia). To measure the distance between HSCs and ECs, two HSCs were randomly picked per scanned image. ImageJ software (the US National Institutes of Health) was used to quantify the indicated areas in scanned images.

### Flow cytometric analysis of BMECs

Femurs of wild-type mice were collected and incubated for 30 min at 37 °C in Dulbecco's modified Eagle's medium (DMEM) containing 1% collagenase D (from *Clostridium histolyticum*, Sigma-Aldrich, catalogue no. C5138-1G), 1 U ml⁻¹ of dispase (Thermo Fisher Scientific, catalogue no. 17105-041) and 1 U ml⁻¹ of DNase (Invitrogen) before cells were dissociated by trituration. Cells were dissociated to generate a single-cell suspension by filtering through a 40-μm nylon mesh. RBCs were eliminated using RBC Lysis Buffer (pluriSelect, Deutscher, catalogue no. 60-00050-11). Cells were subjected to Fc blocking (BioLegend, catalogue no. 101320, 1:200) and stained with FITC-conjugated anti-CD31 (BioLegend, catalogue no. 102506, 1:200), allophycocyanin (APC)-conjugated anti-CD45 (BioLegend, catalogue no. 103112, 1:200), phycoerythrin (PE)-conjugated anti-Ly6a (BioLegend, catalogue no. 122507, 1:200) and APC/Cy7-conjugated anti-Ly6c (BioLegend, catalogue no. 128025, 1:200) antibodies for 40 min. Analysis was performed with an SH800S Cell Sorter (SONY) and FlowJo software (Tree Star).

### ScRNA-seq

ScRNA-seq was performed using a Chromium instrument (10x Genomics). Single-cell suspensions from femurs were generated as described above. Cells were subjected to Fc blocking and stained with FITC-conjugated anti-mouse CD31 (BioLegend, catalogue no. 102506, 1:200) and APC-conjugated anti-mouse CD45 (BioLegend, catalogue no. 103112, 1:200) antibodies for 40 min. CD45⁻CD31⁺ BMECs were sorted with an SH800S Cell Sorter. Libraries were constructed using the commercial microdroplet-based platform Chromium Single Cell 3′ v.3.1 (10x Genomics) according to the manufacturer's instructions. A DNBSEQ-G400 platform (MGI) was used to sequence 28 and 91 bp-end reads. Sequenced reads were aligned to the reference genome (mm10) using Cell Ranger v.3.0.2 software (10x Genomics). The data were processed and analysed using Seurat package v.3.5.2 43. UMAP was used for dimensionality reduction that captures both local and global structures in scRNA-seq data. Clusters were visualized in two dimensions as a feature or violin plot and annotated based on expression of canonical endothelial subtype markers. To construct the lists of upregulated genes in type S ECs, differentially expressed genes were extracted using Wilcoxon's rank-sum test in the Seurat package. IPA (Ingenuity Systems) was performed to identify the most significant pathways and to construct molecular interaction networks of differentially expressed genes (log(FC) > 0.5 or <−0.5) in type S ECs. In brief, the gene list file containing gene symbols, FCs and *P* values was uploaded to IPA and core analysis was performed.

### Flow cytometric analysis of HSPCs and HSCs

Femurs of 12-week-old wild-type mice were collected in PBS. The epiphyses and metaphyses were crushed separately and BM cells were collected for analysis and isolation of HSCs and HSPCs. After haemolysis, surface markers were stained with fluorophore-conjugated antibodies. HSCs and HPCs were phenotypically defined based on the presence and absence of surface markers, as follows: HSCs, lineage⁻Sca-1⁺c-Kit⁺Flt3⁻CD150⁺CD48⁻; HPCs, lineage⁻Sca-1⁺c-Kit⁺Flt3⁺. The methods used to identify and isolate HSCs and other haematopoietic cells have been described previously[32,46,47]. Cell sorting was performed using a FACS Aria II or FACS Aria IIIu instrument (BD Biosciences). Data were analysed with FlowJo software (Tree Star). Anti-lineage marker (CD4, CD8a, Gr-1, Mac-1, Ter-119, B220)-PerCP-Cy5.5, anti-c-Kit-APC-Cy7, anti-Sca-1-PE-Cy7, anti-CD150-PE, anti-CD48-FITC and anti-Flt3-APC

antibodies were used. These antibodies were purchased from BD Biosciences, BioLegend, TONBO Biosciences and eBioscience.

## Quantitative PCR analysis

Total RNA was prepared from isolated cells, and reverse transcription was performed using Superscript III (Invitrogen). The qPCR assays were performed with the ABI 7500 Fast Real-Time PCR System using TaqMan Fast Universal PCR Master Mix (Applied Biosystems) and TaqMan Gene Expression Assay Mix with *Col1a1* (Mm01302043_g1). A mouse β-actin (Mm00607939_s1) assay mix served as an endogenous control. Data were analysed using 7500 Fast System SDS software v.1.3.1.

## Cell-cycle analysis of HSPCs

Femurs from wild-type mice at 12 weeks postnatally were collected in PBS containing 2% fetal calf serum (FCS). The epiphyses and metaphyses were crushed separately using a mortar and pestle on ice followed by filtration with a 70-μm cell strainer (Corning), the diaphyses were flushed with PBS containing 2% FCS and BM cells were collected. After lysis of RBCs, 1–4 million BMMNCs were aliquoted into Eppendorf tubes and 1 μl of Fc-block was added to each sample and incubated for 10 min at 4 °C. Anti-CD150-BV421, anti-CD48-FITC, anti-lineage (CD4, CD8a, Gr-1, Mac-1, Ter-119, B220)-PerCP-Cy5.5, anti-c-Kit-APC-Cy7 and anti-Sca-1-PE-Cy7 antibodies were used to detect surface antigens. Stained samples were centrifuged at 340*g* (2,000 r.p.m.) and 4 °C for 5 min and then 250 μl of BD Cytofix/Cytoperm (BD Biosciences) was added and incubated for 20 min at room temperature for fixation. Fixed cells were washed twice at 2,000 r.p.m. and 4 °C with 1 ml of BD Perm/Wash buffer (BD Biosciences). Each sample was stained with 1 μl of Ki67-eFluor660 (eBioscience) for 30 min at room temperature in the dark. Ki67-stained cells were washed once at 2,000 r.p.m. and 4 °C with BD Perm/Wash buffer. Samples were resuspended in 500 μl of PBS containing 2% FCS and 10 μg ml$^{-1}$ of Hoechst 33432, filtered and then analysed with a BD FACS Aria IIIu instrument.

## BMT

Femurs of wild-type mice at 12 weeks postnatally were collected in PBS containing 2% FCS. The epiphyses and metaphyses were crushed separately and the diaphyses were flushed to collect BM cells for BMT. A total of $4 \times 10^5$ BMMNCs from C57BL/6J-Ly5.2 and $4 \times 10^5$ BMMNCs from a competitor recipient (C57BL/6J-Ly5.1) were transplanted retro-orbitally into lethally irradiated (9.5 Gy) recipients (Ly5.1). At 1, 2, 3 and 4 months after BMT, peripheral blood was collected and the percentage of donor-derived cells and their differentiation status were determined using MACSQuant v.2.1.1 (Miltenyi). Then, 40–80 μl of peripheral blood was sampled from the retro-orbital plexus using heparinized glass capillary tubes (Drummond Scientific) and suspended in 1 ml of PBS containing heparin. The blood suspension was centrifuged at 340*g* for 3 min. The supernatant was discarded and the pellet was resuspended in 1 ml of PBS containing 1.2% (w:v) dextran (200 kDa; Nacalai Tesque) for 45 min at room temperature. The supernatant was centrifuged at 340*g* for 3 min and pellets were resuspended in 0.17 M NH$_4$Cl solution to lyse residual RBCs for 5–10 min until the suspension became clear. Cells were resuspended in 50 μl of PBS containing 2% FCS with 0.3 μl of Fc-block. Surface antigens were stained using the following antibody panel: Gr1-PE-Cy7, Mac-1-PE-Cy7, B220-APC, CD4-PerCP-Cy5.5, CD8a-PerCP-Cy5.5, CD45.1-PE and CD45.2-FITC; 0.3 μl of each antibody was added per sample. The frequency (%) of donor-derived cells was calculated as follows:

100 × donor-derived (GFP$^+$ or Ly5.2$^+$Ly5.1$^-$) cells (%)/(donor-derived cells (%) + competitor- or recipient-derived (Ly5.2-Ly5.1$^+$ or GFP$^-$) cells (%)).

Myeloid cells, B cells and T cells were marked by Gr-1$^+$ or Mac-1$^+$, B220$^+$ and CD4$^+$ or CD8$^+$, respectively. Total cell chimaerism represents the frequency of donor-derived Ly5.2$^+$Ly5.1$^-$ cells over the frequency of Ly5.2$^-$Ly5.1$^+$ cells among mononuclear cells. At 4 months after BMT,

the frequency of donor-derived cells in BM was determined using one femur and tibia per recipient. After counting BM cells using a TC10 automated counter (BioRad), equal volumes of cell suspensions (20–30% of total volume) from each recipient were pooled and $2 \times 10^6$ cells were resuspended in SF-O3 medium containing 0.1% BSA. The cell suspension (200 μl per recipient) was transplanted by retro-orbital injection into lethally (9.5 Gy) irradiated Ly5.1$^+$ recipients with a 1-ml syringe and 27-gauge needle. The remaining cells were stained to assess BM chimaerism. Anti-CD150-BV421, anti-CD48-PE, anti-Flt3-APC, anti-lineage (CD4, CD8a, Gr-1, Mac-1, Ter-119, B220)-PerCP-Cy5.5, anti-c-Kit-APC-Cy7, anti-Sca-1-PE-Cy7, anti-Ly5.1-Alexa Fluor-700 and anti-Ly5.2-FITC antibodies were used to detect surface antigens. Then, 1 μl of each antibody was added per sample.

For limiting dilution analysis to estimate the frequency of HSCs, BM cells were collected from the metaphyseal or diaphyseal region of a cohort of four mice, as described above. HSCs were sorted at a density of 1, 3 or 10 cells per well directly into a 96-well, non-treated, U-bottomed plate (Greiner) containing 300 μl of SF-O3 medium with $8 \times 10^5$ Ly5.1$^+$ whole BM cells. Individual cell suspensions were carefully collected from each well using a 27-gauge, 1-ml syringe (Terumo) and subsequently administered via retro-orbital injection into lethally irradiated (8.5 Gy) Ly5.1$^+$ recipient mice. At 3 months posttransplantation, the frequency of donor-derived cells was assessed as detailed above. Mice exhibiting donor-derived cell frequencies >0.1% and the existence of donor-derived B cells, T cells and myeloid cells were considered to be successfully repopulated. The HSC frequency was estimated using the elda package of R software v.1.5.0 (R-project.org).

## TEM analysis

Bone samples were prepared for TEM observation as described previously[48]. Briefly, the tissues were dissected and fixed with 2.5% glutaraldehyde (TAAB Laboratories Equipment Ltd) diluted in 30 mM HEPES buffer (pH 7.4, Sigma-Aldrich) for 16–24 h under rotation at 4 °C. Decalcification was performed with 0.5 M EDTA for 3 weeks at 4 °C under rotation. After 2 h of postfixation with 1.0% OsO$_4$ (TAAB Laboratories Equipment Ltd), samples were dehydrated in a series of increasing concentrations of ethanol (70–100%) and then in acetone (Sigma-Aldrich), *n*-butyl glycidyl ether (QY1; Oken-shoji Co. Ltd) and graded concentrations of Epon prepared in QY-1. Samples were then incubated in 100% Epon (100 g of Epon contains 27.0 g of 1-methylnicotinamide, 51.3 g of EPOK-812, 21.9 g of dodecenyl succinic anhydride and 1.1 ml of 2,4,6-tris(dimethylaminomethyl) phenol-30, all from Oken-shoji Co. Ltd) for 72 h at 4 °C to enhance infiltration of the resin. After 72 h of polymerization in 100% Epon at 60 °C, semithin sections (1-μm thick) were prepared with a glass knife and stained with toluidine blue. Ultrathin sections (70-nm thick) were prepared using an ultramicrotome (Leica UC7, Leica Biosystems) with a diamond knife. A slice every ten slices was collected on a copper grid and silicon wafer. Ultrathin sections were stained with uranyl acetate and subsequently with lead citrate for 10 min. Dried sections on copper grids were examined with a transmission electron microscope (JEM-1400plus, JEOL).

## X-ray and micro-CT analyses

Soft X-rays were obtained with Softex type CMB2 (Softex Co.) and micro-computed tomography (CT) images were obtained using a micro-CT apparatus (R_mCT2; Rigaku) under 90 kV and 160 μA with 512 projections per 360°. Micro-CT images were analysed using Tri/3DBON software (Ratoc Systems Engineering).

## Whole-mount Alizarin Red staining

Femurs were collected and incubated overnight in both 100% ethanol and 100% acetone at room temperature. Next, samples were placed in 1% KOH overnight with shaking and in 0.005% Alizarin Red (Sigma-Aldrich, A5533) diluted in 1% KOH for 4 h. Finally, samples were

placed in 1% KOH containing 50% glycerol overnight with shaking, imaged and stored in 100% glycerol.

## Mechanical testing

Femurs were obtained immediately after animals were killed, and the femoral epiphyses were cut out. A femoral epiphysis compression test was conducted using a mechanical testing machine (Model TK-252C, Muromachi Kikai Co.). A load was placed at the centre of the bone at a rate of 2 mm min$^{-1}$ until the bone broke. The parameters analysed included stiffness and ultimate force.

## Statistics and reproducibility

Results are expressed as the mean ± s.d. The comparisons between the averages of two groups were evaluated using two-tailed Student's $t$-test. $P$ values < 0.05 were considered statistically significant. For histological analyses, at least three but typically more independent samples were quantified or qualitatively analysed, with each experimental repeat yielding highly similar results. No statistical method was used to predetermine sample size. No data were excluded from the analyses. The experiments were not randomized. PCR genotyping determined group allocation and the investigators were not blinded in this process. Data analysis was performed by different lab members involved in each analysis, to avoid conscious and unconscious bias. No statistical methods were used to predetermine sample sizes, but our sample sizes are similar to those reported in previous publications[21,39]. Data distribution was assumed to be normal but this was not formally tested.

## Reporting summary

Further information on research design is available in the Nature Portfolio Reporting Summary linked to this article.

## Data availability

Sequencing data that support the findings of the present study have been deposited in the Gene Expression Omnibus under accession no. GSE163451. All other data supporting the findings of the present study are available from the corresponding author upon reasonable request. The URL directing the site of Gene Expression Omnibus (GSE163451): https://www.ncbi.nlm.nih.gov/geo/query/acc.cgi?acc=GSE163451. Source data are provided with this paper.

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

## Acknowledgements

We thank H. M. Kronenberg for kindly providing *Osx-Cre$^{ERT2}$* mice and T. Honjo for kindly providing *Rbpj-flox* mice. This work was supported by Grants-in-Aid for Specially Promoted Research from the Ministry of Education, Culture, Sports, Science, and Technology of Japan, all to Y.K. (grant nos. 18H05042, 18K19553 and 19H03397), by AMED-PRIME (grant nos. 21gm6210017h0003 and 22gm6210017h0004 to Y.K.), by JST (Moonshot R&D) (grant no. JPMJMS 2024 to Y.K.), by research grants (all to Y.K.) from the Inamori Foundation, the Kao Foundation for Arts and Culture, the Takeda Science Foundation, the Mochida Memorial Foundation, the Mitsubishi Foundation, the Cell Science Research Foundation, the SENSHIN Medical Research Foundation, the Sumitomo Foundation, the Naito Foundation and the Uehara Memorial Foundation, and by AMED-CREST (grant nos. JP20bm0704042 and JP20gm1210011 to K.T.).

## Author contributions

J.K., K.T. and Y.K. designed the experiments. T.I., H.K., T.S., I.T-N., T.A., T.T., K.M., S.S., J.K. and K.T. performed the experiments. T.I., D.K., T.S., N.F., J.K., K.T. and Y.K. analysed data. K.H., K.I., M.E., T.M. and K.K.K. provided experimental materials. K.M., M.M., M.N., H.O. and K.T. edited the manuscript. T.I. and Y.K. wrote the manuscript.

## Competing interests

The authors declare no competing interests.

## Additional information

**Extended data** is available for this paper at https://doi.org/10.1038/s41556-023-01240-7.

**Correspondence and requests for materials** should be addressed to Keiyo Takubo or Yoshiaki Kubota.

## a

| | Metaphysis | Diaphysis | Epiphysis |
|---|---|---|---|
| Estimated Number of Cells | 843 | 2273 | 155 |
| Mean Reads per cell | 156046 | 106862 | 108442 |
| Median Gene per cell | 2883 | 1626 | 1903 |
| Valid Barcodes | 96.70% | 96.30% | 95.90% |
| Sequencing Saturation | 79.30% | 80.50% | 85.20% |
| Q30 Bases in Barcode | 89.60% | 89.50% | 89.60% |
| Q30 Bases in RNA Read | 89.60% | 89.60% | 87.80% |
| Q30 Bases in UMI | 88.50% | 88.50% | 88.40% |
| Reads Mapped Confidently to Transcriptome | 62.90% | 51.70% | 74.30% |
| Reads Mapped Confidently to Exnic Regions | 66.60% | 55.20% | 77.90% |
| Reads Mapped Confidently to Intronic Regions | 23.50% | 33.90% | 12.90% |
| Reads Mapped Confidently to Intergenic Regions | 3.70% | 4.70% | 2.50% |

## b

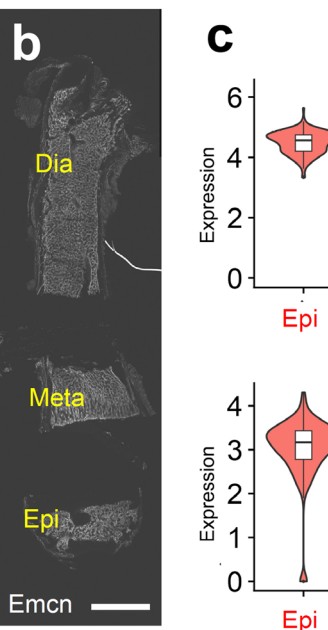

Emcn

## c

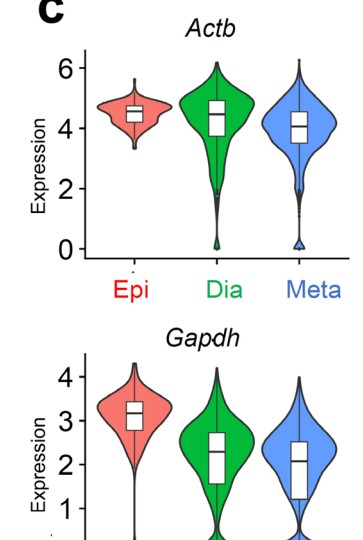

## d

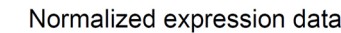

Normalized expression data

Ridge plot for UMI counts

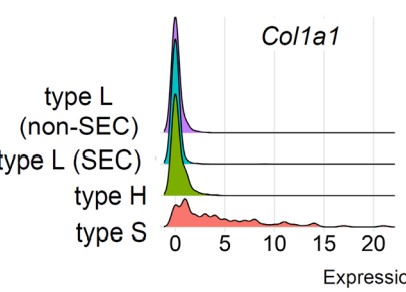

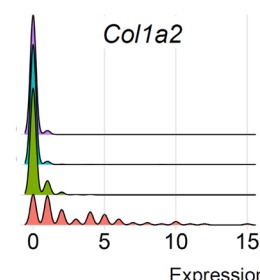

## e    IPA Network analysis

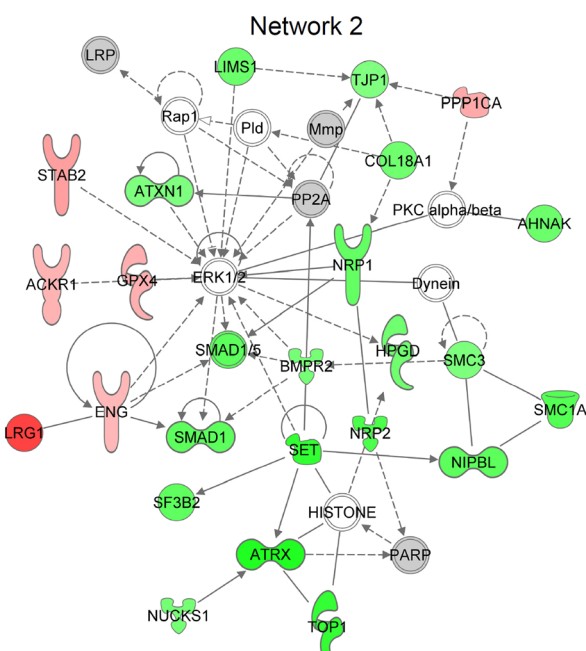

Network 2

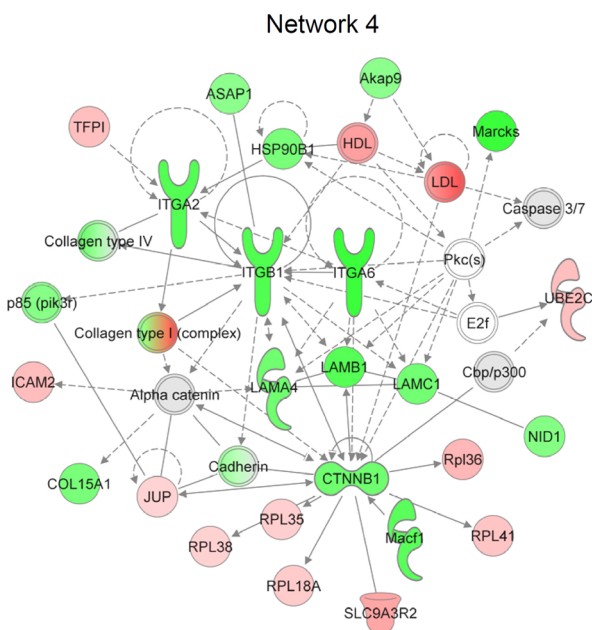

Network 4

**Extended Data Fig. 1 | See next page for caption.**

**Extended Data Fig. 1 | Quality control and characterization of endothelial clusters in scRNA-seq.** (**a**) Statistics of cells analyzed by scRNA-seq. (**b**) Immunohistochemistry of a femur section mechanically dissected into three fragments: diaphysis (Dia), metaphysis (Meta), and epiphysis (Epi). (**c**) Violin plots of internal control genes for the comparison of all fragments. (**d**) A scatter plot of normalized expression and ridge plots of raw UMI (Unique Molecular Identifier) counts for *Col1a1* and *Col1a2*. Both plots confirm that definite populations of type S ECs express these genes abundantly. (**e**) IPA showing the top four networks represented by genes upregulated in type S ECs. Scale bar: 1 mm. In the box plots of **c**, the lower and upper bounds of box represent the 25th and 75th percentile, respectively. The box represents the middle 50% of the data, namely interquartile range (IQR). The horizontal line within the box represents median. The lower and upper bounds of whisker represent the the minimum and maximum values within 1.5 times the interquartile range below the 25th percentile and above the 75th percentile, respectively. The black dots represent outliers. Data presented are the mean ± SD.

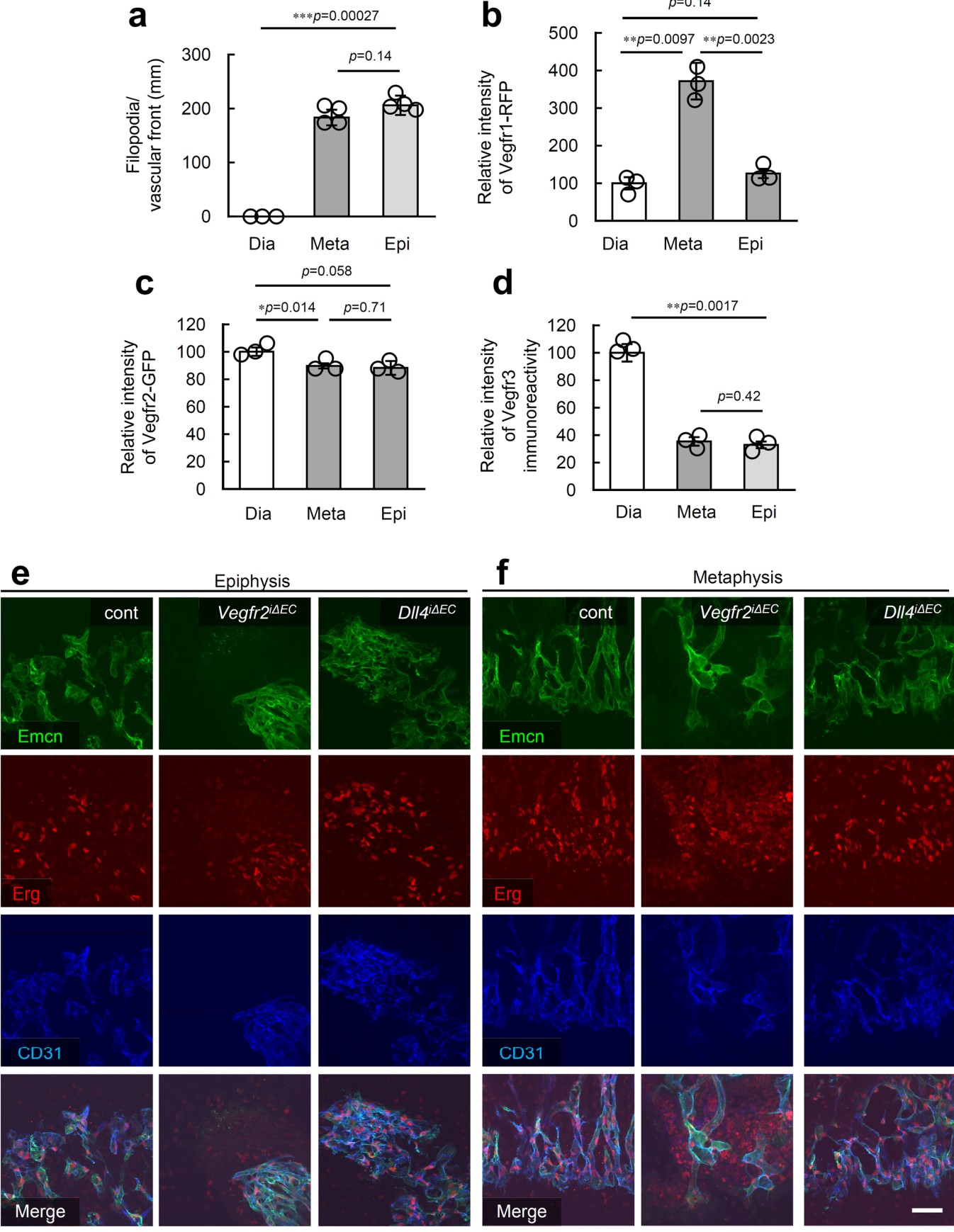

**Extended Data Fig. 2 | See next page for caption.**

**Extended Data Fig. 2 | Expression of VEGF receptors and vascular changes caused by Vegfr2 or Dll4 deficiency.** (**a**) Quantification of the number of tip cell filopodia per 1 mm length of vascular front at P11 (diaphysis: n = 3, metaphysis and epiphysis: n = 4, biologically independent experiments). (**b**–**d**) Quantification of the relative intensities of Vegfr1-RFP, Vegfr2-GFP, and Vegfr3 immunoreactivity at P11 shown in Fig. 2o (n = 3, biologically independent experiments). (**e, f**) Immunohistochemistry of femur sections from mice at P11. ***P<0.001; **P<0.01; *P<0.05. Data presented are the mean ± SD. Scale bar: 50 μm. The comparisons between the averages of two groups were evaluated using the two-tailed Student's t-test.

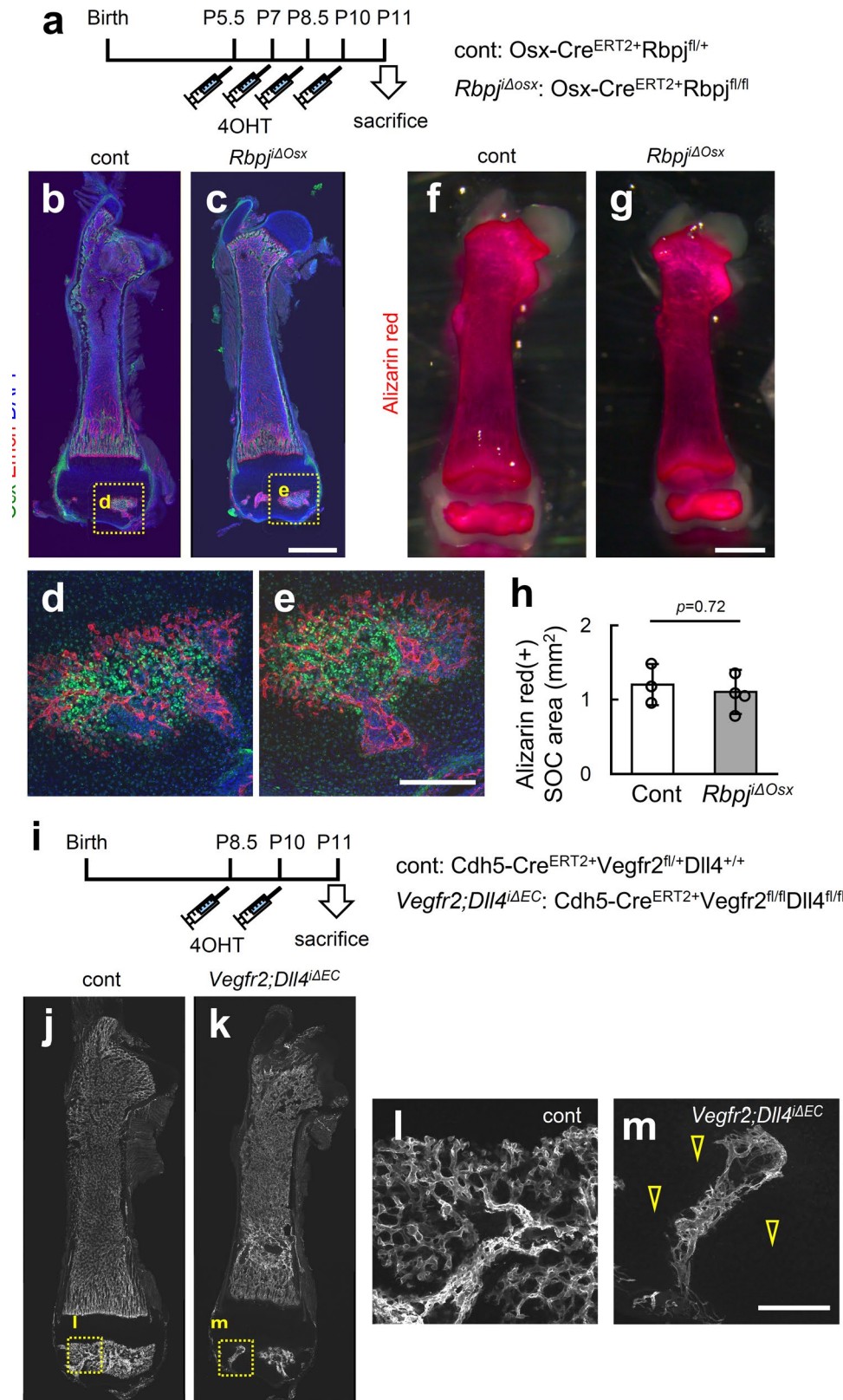

**Extended Data Fig. 3 | Osteogenic Notch signaling and dominance of endothelial Vegfr2 over Dll4. (a)** Protocol for 4OHT injection in neonates. P, postnatal. **(b–h)** Immunohistochemistry of femur sections and whole-mount Alizarin red staining at P11 and quantification (Cont: = 3, *Rbpj*[iΔOsx]: n = 4, biologically independent experiments). **(i)** Protocol for 4OHT injection in neonates. **(j–m)** Immunohistochemistry of femur sections at P11. Combined deletion of *Vegfr2* and *Dll4* in ECs shows that the phenotypes resulting from *Vegfr2* knockout dominate those resulting from *Dll4* knockout (open arrowheads). Scale bars: 1 mm (**b, c, f, g, j, k**); 200 μm (**d, e, l, m**). Data presented are the mean ± SD. The comparisons between the averages of two groups were evaluated using the two-tailed Student's *t*-test.

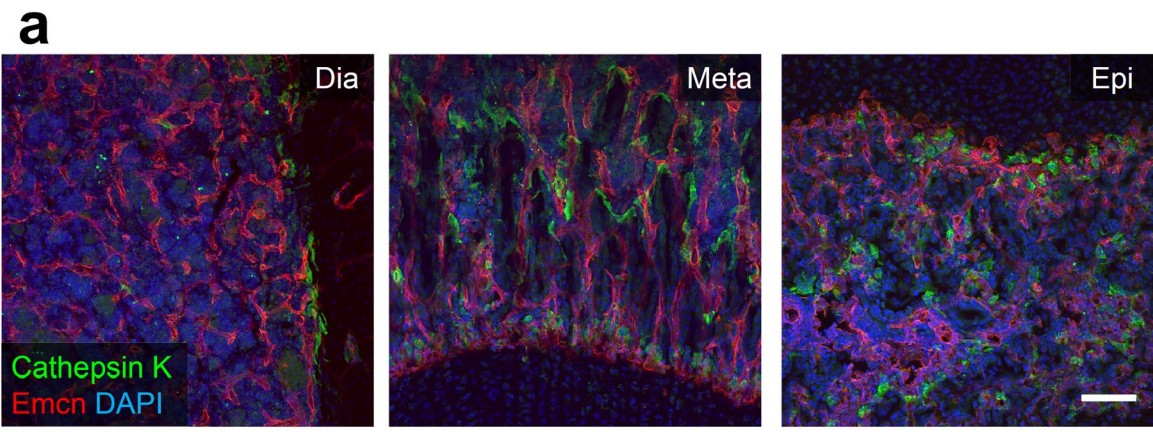

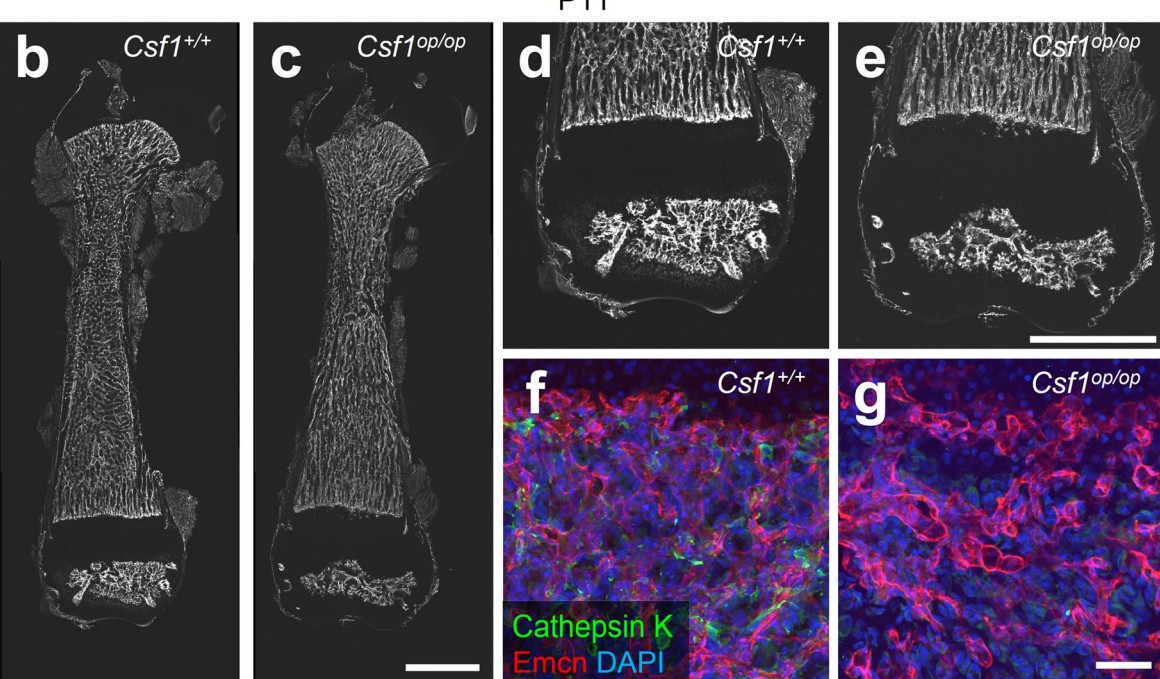

**Extended Data Fig. 4 | Osteoclasts are dispensable for epiphyseal vascularization.** (**a**) Immunohistochemistry of femur sections from mice at P11. (**b–g**) Femur sections at P11. *Csf1op/op* mice lack CatepsinK⁺ osteoclasts, but vessel invasion and formation in the epiphysis occur normally. Scale bars: 1 mm (**b–e**); 50 μm (**a, f, g**).

Epiphysis (*Cxcl12*<sup>+/DsRed</sup>)

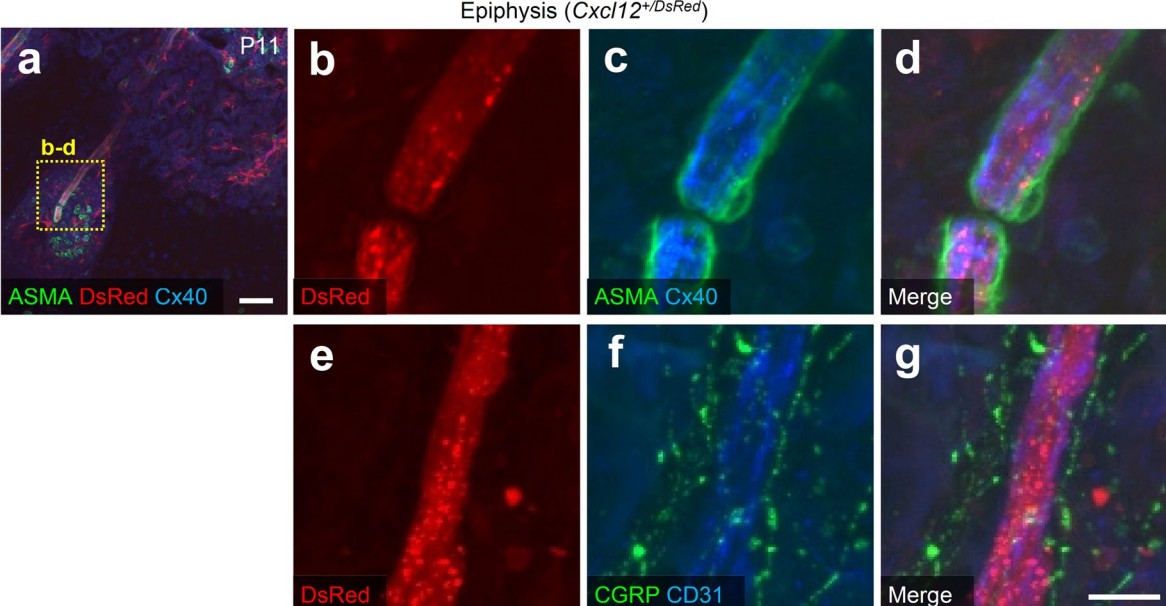

**Extended Data Fig. 5 | DsRed expression in Cxcl12**[+/DsRed] **mouse. (a–g)** Immunohistochemistry of a femur section from a *Cxcl12*[+/DsRed] mouse at P11. DsRed⁺ AECs are covered by ASMA⁺ smooth muscle cells. CGRP⁺ sensory nerves are located around arteries and do not express DsRed. Scale bars: 50 μm (**a**); 20 μm (**b–g**).

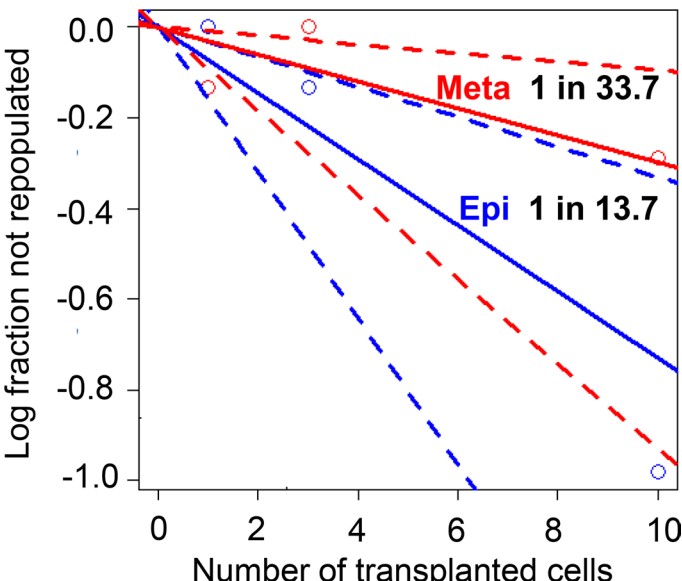

**Extended Data Fig. 6 | Limiting dilution analysis of epiphyseal and metaphyseal cells.** Limiting dilution transplantation analysis using the Poisson statistical method (n = 8 recipients per group). Solid lines indicate the optimal linear model fit, while dotted lines represent the 95% confidence interval. The estimated frequencies of repopulating HSCs within the CD150$^+$CD48$^-$LSK fraction are displayed. Epi, epiphysis; Meta, metaphysis.

# Reporting Summary

## Statistics

For all statistical analyses, confirm that the following items are present in the figure legend, table legend, main text, or Methods section.

| n/a | Confirmed | |
|---|---|---|
| ☐ | ☒ | The exact sample size (*n*) for each experimental group/condition, given as a discrete number and unit of measurement |
| ☐ | ☒ | A statement on whether measurements were taken from distinct samples or whether the same sample was measured repeatedly |
| ☐ | ☒ | The statistical test(s) used AND whether they are one- or two-sided<br>*Only common tests should be described solely by name; describe more complex techniques in the Methods section.* |
| ☒ | ☐ | A description of all covariates tested |
| ☒ | ☐ | A description of any assumptions or corrections, such as tests of normality and adjustment for multiple comparisons |
| ☐ | ☒ | A full description of the statistical parameters including central tendency (e.g. means) or other basic estimates (e.g. regression coefficient) AND variation (e.g. standard deviation) or associated estimates of uncertainty (e.g. confidence intervals) |
| ☐ | ☒ | For null hypothesis testing, the test statistic (e.g. *F*, *t*, *r*) with confidence intervals, effect sizes, degrees of freedom and *P* value noted<br>*Give P values as exact values whenever suitable.* |
| ☒ | ☐ | For Bayesian analysis, information on the choice of priors and Markov chain Monte Carlo settings |
| ☒ | ☐ | For hierarchical and complex designs, identification of the appropriate level for tests and full reporting of outcomes |
| ☒ | ☐ | Estimates of effect sizes (e.g. Cohen's *d*, Pearson's *r*), indicating how they were calculated |

*Our web collection on statistics for biologists contains articles on many of the points above.*

## Software and code

Policy information about availability of computer code

| Data collection | FV10-ASW 3.0 Viewer (Olympus),Strand NGS software v3.3 (Strand Life Sciences), Cell Ranger 3.0 software (10×Genomics) FACS Aria II and FACS Aria IIIu instrument (BD Biosciences). |
|---|---|
| Data analysis | FV10-ASW 3.0 Viewer (Olympus), Image J Software 1.46 (NIH), Seurat package v3.5.2 (Seurat), FlowJo TM v10.8.1(BD Bioscience, elda package of R software ver 1.5.0 (R-project.org), MACSquant ver 2.1.1 (Miltenyi) |

For manuscripts utilizing custom algorithms or software that are central to the research but not yet described in published literature, software must be made available to editors and reviewers. We strongly encourage code deposition in a community repository (e.g. GitHub). See the Nature Portfolio guidelines for submitting code & software for further information.

## Data

Policy information about availability of data

All manuscripts must include a data availability statement. This statement should provide the following information, where applicable:
- Accession codes, unique identifiers, or web links for publicly available datasets
- A description of any restrictions on data availability
- For clinical datasets or third party data, please ensure that the statement adheres to our policy

For scRNAseq data, raw data is available in GEO (GSE163451).  There is no restriction on data availability. Mus musculus (house mouse) genome assembly GRCm38

(mm10) from Genome Reference Consortium Numerical is available in https://www.ncbi.nlm.nih.gov/datasets/genome/GCF_000001635.20/. Numerical source data are provided with this study. All the other data supporting the findings of this study are available from the corresponding author on reasonable request.

## Research involving human participants, their data, or biological material

Policy information about studies with human participants or human data. See also policy information about sex, gender (identity/presentation), and sexual orientation and race, ethnicity and racism.

| | |
|---|---|
| Reporting on sex and gender | N/A |
| Reporting on race, ethnicity, or other socially relevant groupings | N/A |
| Population characteristics | N/A |
| Recruitment | N/A |
| Ethics oversight | N/A |

Note that full information on the approval of the study protocol must also be provided in the manuscript.

# Field-specific reporting

Please select the one below that is the best fit for your research. If you are not sure, read the appropriate sections before making your selection.

☒ Life sciences   ☐ Behavioural & social sciences   ☐ Ecological, evolutionary & environmental sciences

For a reference copy of the document with all sections, see nature.com/documents/nr-reporting-summary-flat.pdf

# Life sciences study design

All studies must disclose on these points even when the disclosure is negative.

| | |
|---|---|
| Sample size | No statistical method was used to calculate sample size. Sample size was determined to be adequate based on the magnitude and consistency of measurable differences between groups. For histological analyses in Figs.1,2, 3, 4, 5,6, and Extended Data Figure 1, 2, 3,4,5,6,7 at least three, but typically more, independent samples, were quantified or qualitatively analysed with each experimental repeat yielding highly similar results. Experimental animals were always compared to litter mate controls. For single cell RNA-seq (Fig. 1,4,5, and Extended Data Figure 1), the sample sizes were the maximum number of cells we could isolate from 20 mice. For Flow cytometry analysis (Fig. 1i), the sample sizes were the maximum number of cells we could isolate from 10 mice . For Flow cytometry analysis (Fig. 7), the sample sizes were the maximum number of cells we could isolate from 6 mice . The number of cells analyzed were sufficient because their numbers were high and isolated from all bones. No statistical methods were used to pre-determine sample sizes but our sample sizes are similar to those reported in previous publications (Okabe et al., 2014; Tai-Nagara et al., 2020). Data distribution was assumed to be normal but this was not formally tested. |
| Data exclusions | No data was excluded. |
| Replication | We have reproduced all our results in a minimum of three independent experimental repeats and using independent samples. Many experiments were repeated independently by two investigators in the lab, in particular experiments shown in Fig. 1, 2, 3, 4, 5, 6,7, and Extended Data Figure 1, 2, 3, 4,5,6. All attempts at replication were successful. |
| Randomization | Randomization was not required for our mouse studies because we determined the genotype using genomic PCR before the experiments. |
| Blinding | PCR genotyping determined group allocation, and the investigators were not blinded in this process. Data analysis was performed by different lab members involved in each analysis, to avoid conscious and unconscious bias. |

# Reporting for specific materials, systems and methods

We require information from authors about some types of materials, experimental systems and methods used in many studies. Here, indicate whether each material, system or method listed is relevant to your study. If you are not sure if a list item applies to your research, read the appropriate section before selecting a response.

## Materials & experimental systems

| n/a | Involved in the study |
|-----|----------------------|
| ☐ | ☒ Antibodies |
| ☒ | ☐ Eukaryotic cell lines |
| ☒ | ☐ Palaeontology and archaeology |
| ☐ | ☒ Animals and other organisms |
| ☒ | ☐ Clinical data |
| ☒ | ☐ Dual use research of concern |
| ☒ | ☐ Plants |

## Methods

| n/a | Involved in the study |
|-----|----------------------|
| ☒ | ☐ ChIP-seq |
| ☐ | ☒ Flow cytometry |
| ☒ | ☐ MRI-based neuroimaging |

## Antibodies

| | |
|---|---|
| Antibodies used | The primary monoclonal antibodies used were, CD31 (Abcam; Cambridge, UK, ab119341; 1:1000), Runx2 (Abcam; ab192256; 1:500), ASMA (Sigma–Aldrich, Saint Louis, MO, USA, A5228; 1:200), Ly6a (BD; 553333; 1:500), anti-CD150 (BioLegend, San Diego, CA, USA; 115905; 1:200), and Endomucin (Santa Cruz; Santa Cruz, CA, USA; sc-65495; 1:500). The primary polyclonal antibodies used were as follows: GFP-Alexa Fluor 488-conjugated (Molecular Probes, Eugene, OR, USA, A21311; 1:500), c-Kit (R&D Systems, Minneapolis, MN, USA, AF1356; 1:500), Sp7/Osterix (Abcam; ab22552; 1:500), Cathepsin K (Abcam; ab19027; 1:500), VEGFR3 (R&D; AF743; 1:1000), CGRP (SIGMA; C8198; 1:8000), and anti-Erg (Abcam, ab92513, 1:2,000). Secondary antibodies used were Alexa Fluor 488-conjugated IgGs (Molecular Probes, A11034, A11006, A11055; 1:500) or Cy3/Cy5 DyLight549/DyeLight649-conjugated IgGs (Jackson ImmunoResearch, West Grove, PA, USA, 711-165-152, 112-165-167, 127-165-160, 711-605-152, 112-605-167, 127-605-160; 1:500). For nuclear staining, specimens were treated with 4',6-diamidino-2-phenylindole (DAPI; Molecular Probes, D-1306). For Flow cytometry analysis, we used the following antibodies. FITC-conjugated anti-CD31 antibodies (Biolegend; 102506; 1:200), APC-conjugated anti-CD45 antibodies (Biolegend; 103112; 1:200), PE-conjugated anti-Ly6a antibodies(Biolegend; 122507; 1:200), and APC/Cy7-800 conjugated anti Ly6c antibodies(Biolegend; 128025; 1:200) |
| Validation | All antibodies were sold by the manufacturer with validation data and citations, and they detected the specified targets in our study as expected. Specificity was confirmed by obtaining the expected pattern of tissue staining with the respective antibodies. <br><br> All commercially available antibodies are routinely tested by he manufacturers. <br> -CD31 (Abcam; Cambridge, UK, ab119341; 1:1000). It has been successfully used for IF in the mouse mesentery by Deng Muller's group (doi: 10.1084/jem.179.3.1059). <br> -Runx2 (Abcam; ab192256; 1:500). It has been successfully used for IF in the mouse dura by Elizabeth Engles's group (doi: 10.1016/j.devcel.2017.07.027.). <br> -ASMA (Sigma–Aldrich, Saint Louis, MO, USA, A5228; 1:200). It has been successfully used for IF in the mouse carotid artery by Adria Giacca's group (doi: 10.1161/ATVBAHA.109.185447). <br> -Ly6a (BD; 553333; 1:500). It has been successfully used for IF in the mouse bone marrow by Ralf Adams's group (doi:10.1038/nature17638). <br> -anti-CD150 (BioLegend, San Diego, CA, USA; 115905; 1:200). It has been successfully used for IF in the mouse bone marrow by Paul Frenette's group (doi: 10.1038/nature12612.). <br> -Endomucin (Santa Cruz; Santa Cruz, CA, USA; sc-65495; 1:500).  It has been successfully used for IF in the mouse bone marrow by Ralf Adams's group (doi:10.1038/nature17638). <br> -GFP-Alexa Fluor 488-conjugated (Molecular Probes, Eugene, OR, USA, A21311; 1:500).  It has been successfully used for IF in the mouse bone marrow by Ralf Adams's group (doi:10.1038/nature17638). <br> -c-Kit (R&D Systems, Minneapolis, MN, USA, AF1356; 1:500).  It has been successfully used for IF in the mouse bone marrow by Paul Frenette's group (doi: 10.1038/nature12612.). <br> -Sp7/Osterix (Abcam; ab22552; 1:500). It has been successfully used for IF in the mouse bone marrow by Ralf Adams's group (doi:10.1038/nature13145). <br> -Cathepsin K (Abcam; ab19027; 1:500). It has been successfully used for IF in the mouse digits by Ken Muneoka's group (doi: 10.1016/j.ydbio.2022.03.007). <br> -VEGFR3 (R&D; AF743; 1:1000), CGRP (SIGMA; C8198; 1:8000). It has been successfully used for IF in the mouse bone marrow by Ralf Adams's group (doi:10.1038/nature13146). <br> -anti-Erg (Abcam, ab92513, 1:2,000).  It has been successfully used for IF in the mouse lung by Ralf Adams's group (doi:10.1038/s41467-017-01738-3). <br> -FITC-conjugated anti-CD31 antibodies (Biolegend; 102506; 1:200). It has been successfully used for flowcytometry in the mouse bone marrow by Ralf Adams's group (doi:10.1038/nature17638). <br> -APC-conjugated anti-CD45 antibodies (Biolegend; 103112; 1:200). It has been successfully used for flowcytometry in the mouse bone marrow by Ralf Adams's group (doi:10.1038/nature17638). <br> -PE-conjugated anti-Ly6a antibodies (Biolegend; 122507; 1:200). It has been successfully used for flowcytometry in the mouse bone marrow by Rubinsztein DC's group (10.1038/ncomms10533). <br> -APC/Cy7-800 conjugated anti Ly6c antibodies (Biolegend; 128025; 1:200). It has been successfully used for flowcytometry in the mouse bone marrow by Rubinsztein DC's group (10.1038/ncomms10533). |

# Animals and other research organisms

Policy information about <u>studies involving animals</u>; <u>ARRIVE guidelines</u> recommended for reporting animal research, and <u>Sex and Gender in Research</u>

| | |
|---|---|
| Laboratory animals | The strains (Cdh5-BAC-CreERT2, Osx-CreERT2,FVegfr2-flox, Dll4-flox, Col1a1-flox,CAG- LSL-EGFP, Vegfr1-BAC-DsRed ,Vegfr2-BAC-GFP, Rbpj-flox, Csf1op/op,Cxcl12+/DsRed) have all been published previously and are described in the methods section with appropriate references. Mice at P7, P11, P13, P18, P28, P35, 3Mo, 4Mo, and 6Mo. Ages of mice used in each experiment are specified in the figures and figure legends. Mouse cages were well ventilated, softly lit and subject to a 12h light/dark cycle. The relative humidity were kept at 45 to 65%. Mouse rooms and cages were kept at a temperature range of 20-24oC. |
| Wild animals | The study did not involve wild animals. |
| Reporting on sex | The vast majority of work was carried out with neonatal materials of unknown sex, but, based on Mendelian genetics, all material would be comprised of equal proportions of male and female tissues, as none of the strains we have used have reported sex-related lethality that would eliminate one or the other sex from litters. The sex of the tissues has been recorded and can be provided if required, but is considered not relevant in the context of our study. |
| Field-collected samples | The study did not involve samples collected from the field. |
| Ethics oversight | Animal experiments were approved by the Institutional Animal Care and Use Committee of Keio University (protocol number A2021-003)  and were performed in accordance with the Guidelines of Keio University for Animal and Recombinant DNA Experiments (protocol number D2005-044). |

Note that full information on the approval of the study protocol must also be provided in the manuscript.

# Flow Cytometry

## Plots

Confirm that:

☒ The axis labels state the marker and fluorochrome used (e.g. CD4-FITC).

☐ The axis scales are clearly visible. Include numbers along axes only for bottom left plot of group (a 'group' is an analysis of identical markers).

☒ All plots are contour plots with outliers or pseudocolor plots.

☒ A numerical value for number of cells or percentage (with statistics) is provided.

## Methodology

| | |
|---|---|
| Sample preparation | Femurs from wild-type mice at postnatal 12 weeks were collected in PBS+2%FCS.Epiphysis, and metaphysis were crushed separately, and diaphysis was flushed to collect bone marrow cells |
| Instrument | Cell sorting was performed using a FACS Aria II, FACS Aria IIIu instrument (BD Biosciences).or SH800S (SONY) |
| Software | Data were analyzed with FlowJo software (Treestar) |
| Cell population abundance | Post-sorted cells were immediately re-analyzed to confirm the validity and purity (>95%) of each sorting. |
| Gating strategy | Debris and dead cells were excluded by forward and side scatter and a negative gate for propidium iodide staining. Percentages of cell doublets/aggregates were checked by FSC-A vs. FSC-W, and SSC-A vs. SSC-W plots. In all FACS studies, we routinely included unstained control samples and compensation tubes. Using these negative and positive control tubes, we set fluorescence voltages and the compensation matrix according to the instructions of the manufacturer. |

☒ Tick this box to confirm that a figure exemplifying the gating strategy is provided in the Supplementary Information.

