## [Peer Review File · Nature Cell Biology]

Peer Review Information

Journal: Nature Cell Biology

Manuscript Title: Spatial heterogeneity of bone marrow endothelial cells unveils a distinct subtype in the epiphysis

Corresponding author name(s): Professor Yoshiaki Kubota

Editorial Notes:

Reviewer Comments & Decisions:

Decision Letter, initial version:

*Please delete the link to your author homepage if you wish to forward this email to co-authors.

Dear Yoshiaki,

Your manuscript, "Spatial heterogeneity of bone marrow endothelial cells unveils a distinct capillary subtype in the epiphysis", has now been seen by 3 referees, who are experts in bone biology and vasculature, scRNAseq (referee 1); bone marrow, endothelial cells, HSC niche, scRNAseq (referee 2); and bone marrow endothelial cells, HSC niche (referee 3). As you will see from their comments (attached below) they find this work of potential interest, but have raised substantial concerns, which in our view would need to be addressed with considerable revisions before we can consider publication

in Nature Cell Biology.

Nature Cell Biology editors discuss the referee reports in detail within the editorial team, including the chief editor, to identify key referee points that should be addressed with priority, and requests that are overruled as being beyond the scope of the current study. To guide the scope of the revisions, I have listed these points below. I should stress that the referees' concerns point to a premature dataset and these points would need to be addressed with experiments and data, and reconsideration of the study for this journal and re-engagement of referees would depend on strength of these revisions.

In particular, it would be essential to:

(A) Perform additional experiments to develop the dataset and bolster the claims in the study with regard to HSCs, as indicated by:

Referee #2:

"Type S ECs have low levels of Kit-ligand but high levels of Tgfb β s that might indicate this specific vascular niche is primarily delegated with supporting the HSPCs that are quiescent rather than undergoing self-renewal or expansion. While the authors show repopulating HSPCs reside in vicinity of the Type S ECs, the unique attributes of these HSPCs are unclear. For example, do Type S ECs prevent excessive differentiation of the HSPCs to osteoclastic cells? Do HSPCs could be mobilized from the Type S zoned region to the peripheral circulation?"

Referee #3:

"Figure 5. Figure 5a-g: c-kit $^+$ expression by itself is suboptimal to identify hematopoietic stem/progenitor cells. If the authors wish to highlight the anatomic association of hematopoietic stem/progenitor cells with vessels in the post natal epiphysis, then additional markers should be added, to include Sca-1 and additional lineage markers so that at least ckit $^+$ sca-1 $^+$ lin $^-$ cells can be identified in the epiphyses. Several labs have demonstrated staining for hematopoietic stem cells in situ using various markers, such as CD150+CD48/41 $^-$ (Kunisaki et al. Nature 2013;502:637) and the authors could follow any of these approaches to provide more convincing data regarding hematopoietic stem/progenitor association with vessels in the epiphysis post-natally.

The same concern noted above applies for panels 5k-p. If the authors are solely going to utilize ckit $^+$ to identify hematopoietic progenitors, then the text needs to be edited to reflect that limitation".

"If the percentages of phenotypic HSCs or long term HSCs are not different in the epiphysis versus the metaphysis, but the authors have observed a difference in HSC repopulation in transplanted mice, does this mean that the differences in donor cell repopulation in recipient mice are due to non-HSC effects, i.e. adventitial or supportive cells in the BM grafts? Were these transplants competitive? If not, they should be repeated using competitor cells to rigorously compare the HSC repopulating capacity of the donor hematopoietic cells from the epiphysis vs. the metaphysis.

One way to resolve the concern regarding non-HSC effects causing the differences in donor cell engraftment in recipient mice would be to FACS sort HSCs (i.e. CD150+CD48 $^-$ lin $^-$ BM cells) and transplant at limiting dilution, along with competitor BM cells into recipient mice since this would

isolate comparison to the HSC populations in the donor mice".

"Figure 6f. Does the diminution in donor engraftment in the epiphyseal donor BM cell – transplanted secondary mice reflect that both the epiphysis and the metaphysis contain relatively few long term HSCs compared to the diaphyseal portion of the mouse femur? (Acar et al. Nature 2015;526:126)".

"It has been suggested that quiescent HSCs reside in relative proximity to certain types of bone marrow ECs (e.g. arterioles), whereas more proliferative HSCs reside in different EC niches (Kunisaki et al. Nature 2013;502:637). Have the authors interrogated the cell cycle status of HSCs residing close to the type S epiphyseal ECs/vessels? This would be important information to add to the manuscript".

(B) Further strengthen claims related to the newly identified type S ECs and provide mechanistic insights as requested by:

Referee #1:

"Are the cell surface markers of type S ECs overlapped and exchangeable to H type ECs during development?"

"It was mentioned that type S ECs atypically contributed to the acquisition of bone strength by secreting type I collagen. Recent studies (for example, PMID: 32483430) have found that endothelial cells produce many angiocrine factors to regulate bone and cartilage via versatile mechanisms, including Hedgehog, Notch, WNT, bone morphogenetic protein (BMP), fibroblast growth factor (FGF), insulin-like growth factor (IGF), and platelet-derived growth factor (PDGF). It would be relevant to discuss this possibility by referring to this study. It would be informative to discuss if type S ECs might also produce other potential angiocrine factors with osteogenic property to regulate bone and cartilage, which would enhance the overview and potential of this study".

Referee #2:

"The demonstration that mesenchymal cell-derived type I collagen is insufficient for epiphysial osteogenesis, but that Type S EC-derived complement this insufficiency by secreting type I collagen is probably the most significant finding in this manuscript. This is important because to date no other tissue-specific ECs have been shown to express Col1a1,2 type collagens and the majority of the collagens produced by ECs are Collagen Type IV or other types of collagen. To prove this point, the authors have generated endothelial-specific Col1a1-knockout (Cdh5-BAC-CreERT2+Col1a1flox/flox, referred to as Col1a1 Δ EC) mice, that manifested impaired osteogenesis. To further substantiate this important finding, beyond shadow of doubt, the authors might want to purify Type S ECs and show how these cells can synthesize Col1a1 or 2 as well as Collagen type IV or other types as well. The mechanism by which Type S ECs could produce Col1a1 or 2 and balance production of other collagen types should also be investigated".

"In figure 1, one can observe that as compared to Type L and Type H ECs, the number of Type S ECs in cluster 8 compose of a diminishingly a very small percentage of total ECs. In this regard, there

could be under representation or drop out of certain transcriptomes in these scant population of Type S ECs. In addition, the FKPM values for Col1a1 and Cola2 as well as Cxcl12, Tgfb1, Tgfb2 and osteopontin is not well defined. If possible, the authors might consider performing bulk RNA sequencing to acquire a better understanding of the level of expression of these factors in Type S ECs".

Referee #3:

"Figure 1. In Figure 1C, it appears in the UMAP that there is potential overlap between the "type S" ECs and the arteriole ECs. Also the arteriole ECs have high expression of Ly6a, which is noted to be high in the Type S ECs. Are the newly described type S ECs overlapping or redundant to the arteriolar ECs that others have previously characterized?"

(C) All other referee concerns pertaining to performing quantifications, strengthening existing data, providing clarifications and textual changes, should also be addressed.

(D) Finally please pay close attention to our guidelines on statistical and methodological reporting (listed below) as failure to do so may delay the reconsideration of the revised manuscript. In particular please provide:

- a Supplementary Figure including unprocessed images of all gels/blots in the form of a multi-page pdf file. Please ensure that blots/gels are labeled and the sections presented in the figures are clearly indicated.
- a Supplementary Table including all numerical source data in Excel format, with data for different figures provided as different sheets within a single Excel file. The file should include source data giving rise to graphical representations and statistical descriptions in the paper and for all instances where the figures present representative experiments of multiple independent repeats, the source data of all repeats should be provided.

We would be happy to consider a revised manuscript that would satisfactorily address these points, unless a similar paper is published elsewhere, or is accepted for publication in Nature Cell Biology in the meantime.

- ensure that it conforms to our format instructions and publication policies (see below and www.nature.com/nature/authors/).
- provide a point-by-point rebuttal to the full referee reports verbatim, as provided at the end of this letter.
- provide the completed Editorial Policy Checklist (found here <https://www.nature.com/authors/policies/Policy.pdf>), and Reporting Summary (found

here <https://www.nature.com/authors/policies/ReportingSummary.pdf>). This is essential for reconsideration of the manuscript and these documents will be available to editors and referees in the event of peer review. For more information see <http://www.nature.com/authors/policies/availability.html> or contact me.

Nature Cell Biology is committed to improving transparency in authorship. As part of our efforts in this direction, we are now requesting that all authors identified as 'corresponding author' on published papers create and link their Open Researcher and Contributor Identifier (ORCID) with their account on the Manuscript Tracking System (MTS), prior to acceptance. ORCID helps the scientific community achieve unambiguous attribution of all scholarly contributions. You can create and link your ORCID from the home page of the MTS by clicking on 'Modify my Springer Nature account'. For more information please visit www.springernature.com/orcid.

[Redacted]

We would like to receive a revised submission within six months. We would be happy to consider a revision even after this timeframe, however if the resubmission deadline is missed and the paper is eventually published, the submission date will be the date when the revised manuscript was received.

We hope that you will find our referees' comments, and editorial guidance helpful. Please do not hesitate to contact me if there is anything you would like to discuss.

Best wishes,

Stelios

Stylios Lefkopoulos, PhD
He/him/his
Associate Editor
Nature Cell Biology
Springer Nature
Heidelberger Platz 3, 14197 Berlin, Germany

E-mail: stylios.lefkopoulos@springernature.com

Twitter: @s_lefkopoulos

Reviewers' Comments:

Reviewer #1:

Remarks to the Author:

The paper regarding Spatial heterogeneity of bone marrow endothelial cells unveils a distinct capillary subtype in the epiphysis is well presented. The authors showed that epiphysis consists of a spatial heterogeneity of BMECs, with features of capillary endothelial cells (ECs), termed type S (secondary ossification) ECs. They carried out genetic KO of VEGFR and Dll4 to investigate the bone phenotype and angiogenic features. They showed that type S ECs produce type I collagen, and modulate bone and hematopoietic development. This is an interesting study.

There are some possible issues

Are the cell surface markers of type S ECs overlapped and exchangeable to H type ECs during development ?

Figure 2 i. Can you provide some quantitative measurement instead of single %, how many times were done? N=?

Figure 2, H-P. Can you provide some quantitative measurement for these described features?

Figure 3, There are three main subtypes of VEGFR, numbered 1, 2 and 3. Is VEGFR2 mainly expressed in epiphysis, relative to VEGFR1 and 3 ? What are the proportion of VEGFR1, 2, and 3 expression in epiphysis and metaphysis? What might happen if VEGFR1 and 3 were deleted?

Figure 3, There are four main subtypes of Delta-like protein, numbered 1, 2, 3 and 4. Why was Dll4 used to be chosen to delete? Is Dll4 mainly expressed in epiphysis, relative to other subtypes ? What are the proportion of Dll1-4 expression in epiphysis and metaphysis? What might happen if Dll like Dll1-3 were deleted?

It was mentioned that type S ECs atypically contributed to the acquisition of bone strength by secreting type I collagen. Recent studies (for example, PMID: 32483430) have found that endothelial cells produce many angiocrine factors to regulate bone and cartilage via versatile mechanisms, including Hedgehog, Notch, WNT, bone morphogenetic protein (BMP), fibroblast growth factor (FGF), insulin-like growth factor (IGF), and platelet-derived growth factor (PDGF). It would be relevant to discuss this possibility by referring to this study. It would be informative to discuss if type S ECs might also produce other potential angiocrine factors with osteogenic property to regulate bone and cartilage, which would enhance the overview and potential of this study.

Figure 4 d-q, can you provide some quantitative measurement for these described features?

Figure 5 a-g, can you provide some quantitative measurement for these described features?

Figure 5 k-p, can you provide some quantitative measurement for these described features?

It is not sure if type S ECs were observed in humans? It would be informative to show where type S ECs expressed in human bone?

Reviewer #2:

Remarks to the Author:

In this study, the authors profiled bone marrow endothelial cells (ECs) isolated from mouse femurs, carefully segregating epiphysis, diaphysis and metaphysis regions of the mouse femur. Using scRNA-seq they identified a previously unappreciated capillary subtype, within the epiphysis termed type S ECs. These ECs manifested unique phenotypic signature $Ly6a^{high}Ly6c^{high}$ and expressed *Col1a1* and *Col1a2* which are commonly expressed by the non-vascular cells. The authors also demonstrated that repopulating HSPCs reside within epiphysis as well. Indeed, Type S ECs supported hematopoiesis by presumably secreting *Cxcl12*, *Tgfb1*, *Tgfb2*, and *Spp1* (Osteopontin). They also used genetic models in which the expression of *Col1a1*, *VEGFR2*, and *Dll4* among others were deleted in the adult endothelial cells. These studies demonstrated that Type S cells are delegated with supporting/balancing hematopoiesis and osteogenesis promoting bone growth/remodeling.

Comments:

The contribution of the vessels in the epiphysis has been neglected as this portion of the bone was often discarded. Therefore, the angiocrine contribution of these unique endothelial cells (ECs) to balance of bone formation and hematopoiesis was not well studied. In this regard, the data presented in this paper that assigns an operational, phenotypic and functional unique attributes to Type S epiphyseal vessels are novel and could have both mechanistic and translational value. The finding that Type S epiphyseal ECs could deposit *Col1a1* and *Col1a2* as well as supply angiocrine factors *Cxcl12*, *Tgfb1*, *Tgfb2* and osteopontin are based on genetic knock down of these factors specifically in the adult ECs and as such are convincing. Overall, the findings by the authors that unique population of ECs, namely Type S dendritic shaped vessels arborize the epiphyses is novel and can open the door for therapeutic approaches to study hematopoiesis and osteogenesis.

Specific concerns:

1) The demonstration that mesenchymal cell-derived type I collagen is insufficient for epiphyseal osteogenesis, but that Type S EC-derived complement this insufficiency by secreting type I collagen is probably the most significant finding in this manuscript. This is important because to date no other tissue-specific ECs have been shown to express *Col1a1,2* type collagens and the majority of the collagens produced by ECs are Collagen Type IV or other types of collagen. To prove this point, the authors have generated endothelial-specific *Col1a1*-knockout (*Cdh5-BAC-CreERT2+Col1a1^{flox/flox}*, referred to as *Col1a1^{ΔEC}*) mice, that manifested impaired osteogenesis. To further substantiate this important finding, beyond shadow of doubt, the authors might want to purify Type S ECs and show how these cells can synthesize *Col1a1* or 2 as well as Collagen type IV or other types as well. The mechanism by which Type S ECs could produce *Col1a1* or 2 and balance production of other collagen types should also be investigated.

2) Can the authors provide a schema in which Type S ECs sense hematopoietic mass and as such balance osteogenic activity as well? For example, is this balance is regulated by differential activation of osteoblasts or osteoclasts? What is thermostat within the Type S ECs or their microenvironment that balance hematopoiesis and osteogenesis?

3) In figure 1, one can observe that as compared to Type L and Type H ECs, the number of Type S ECs in cluster 8 compose of a diminishingly a very small percentage of total ECs. In this regard, there could be under representation or drop out of certain transcriptomes in these scant population of Type

S ECs. In addition, the FKPM values for Col1a1 and Cola2 as well as Cxcl12, Tgfb1, Tgfb2 and osteopontin is not well defined. If possible, the authors might consider performing bulk RNA sequencing to acquire a better understanding of the level of expression of these factors in Type S ECs.

4) Type S ECs have low levels of Kit-ligand but high levels of Tgfb2s that might indicate this specific vascular niche is primarily delegated with supporting the HSPCs that are quiescent rather than undergoing self-renewal or expansion. While the authors show repopulating HSPCs reside in vicinity of the Type S ECs, the unique attributes of these HSPCs are unclear. For example, do Type S ECs prevent excessive differentiation of the HSPCs to osteoclastic cells? Do HSPCs could be mobilized from the Type S zoned region to the peripheral circulation?

Reviewer #3:

Remarks to the Author:

In this manuscript by Iga et al., the authors describe a subtype of bone marrow endothelial cells, termed type S ECs, in the epiphysis, which they demonstrate to have a role in regulating local ossification and providing a niche for HSCs. There are several interesting findings in the manuscript, but I have concerns about some of the analyses performed, particularly as relates to the characterization of the hematopoietic effects.

Figure 1. In Figure 1C, it appears in the UMAP that there is potential overlap between the "type S" ECs and the arteriole ECs. Also the arteriole ECs have high expression of Ly6a, which is noted to be high in the Type S ECs. Are the newly described type S ECs overlapping or redundant to the arteriolar ECs that others have previously characterized?

Figure 2. The images in Figure 2O are qualitatively interesting, but would be improved if expression of the VEGFR receptors was quantified. Authors should be able to perform flow cytometry on ECs from the different regions to quantify the different VEGFR isoforms on the different EC subsets.

The labeling in panel 2O is confusing. Any abbreviations shown in the figure panels should be spelled out in the Figure legend for that figure panel.

Figure 3. This panel elegantly displays the effects of inducible, EC-specific deletion of Vegfr2 or Dll4 on bone formation in the epiphysis and metaphysis via detection using Alizarin red and Osx+ cells. However, it would help to see the primary effects in high power on the epiphyseal and metaphyseal blood vessels at the same time points. Can the authors utilize additional EC markers to illuminate the vasculature at the same time points in each group and show in higher power? The low power views of endomucin+ vessels/vascular structures in the epiphysis in panel 3M are not satisfactory.

Figure 4. The differences in controls and the Col1a1delta-EC are subtle and not readily discerned by the images shown in panels l vs. m, n vs. o and p vs. q. It would be helpful to show higher magnification views to demonstrate the differences the authors are trying to show. Otherwise, the text should be qualified to reflect that the differences in these mice are modest and conclusions should be only carefully drawn from these data.

Figure 5. Figure 5a-g: c-kit+ expression by itself is suboptimal to identify hematopoietic stem/progenitor cells. If the authors wish to highlight the anatomic association of hematopoietic

stem/progenitor cells with vessels in the post natal epiphysis, then additional markers should be added, to include Sca-1 and additional lineage markers so that at least $ckit+sca-1+lin^-$ cells can be identified in the epiphyses. Several labs have demonstrated staining for hematopoietic stem cells in situ using various markers, such as $CD150+CD48/41^-$ (Kunisaki et al. Nature 2013;502:637) and the authors could follow any of these approaches to provide more convincing data regarding hematopoietic stem/progenitor association with vessels in the epiphysis post-natally.

The same concern noted above applies for panels 5k-p. If the authors are solely going to utilize $ckit+$ to identify hematopoietic progenitors, then the text needs to be edited to reflect that limitation.

Figure 5x. Please show representative flow cytometric plot of the 34-LSK cells in the control and $Vegfr2\Delta$ EC group.

Figure 6. Representative percentages of cell populations should be shown for each cell population gate shown in 6a.

6b. If the percentages of phenotypic HSCs or long term HSCs are not different in the epiphysis versus the metaphysis, but the authors have observed a difference in HSC repopulation in transplanted mice, does this mean that the differences in donor cell repopulation in recipient mice are due to non-HSC effects, i.e. adventitial or supportive cells in the BM grafts? Were these transplants competitive? If not, they should be repeated using competitor cells to rigorously compare the HSC repopulating capacity of the donor hematopoietic cells from the epiphysis vs. the metaphysis.

One way to resolve the concern regarding non-HSC effects causing the differences in donor cell engraftment in recipient mice would be to FACS sort HSCs (i.e. $CD150+CD48-lin^-$ BM cells) and transplant at limiting dilution, along with competitor BM cells into recipient mice since this would isolate comparison to the HSC populations in the donor mice.

Figure 6f. Does the diminution in donor engraftment in the epiphyseal donor BM cell – transplanted secondary mice reflect that both the epiphysis and the metaphysis contain relatively few long term HSCs compared to the diaphyseal portion of the mouse femur? (Acar et al. Nature 2015;526:126).

It has been suggested that quiescent HSCs reside in relative proximity to certain types of bone marrow ECs (e.g. arterioles), whereas more proliferative HSCs reside in different EC niches (Kunisaki et al. Nature 2013;502:637). Have the authors interrogated the cell cycle status of HSCs residing close to the type S epiphyseal ECs/vessels? This would be important information to add to the manuscript.

REFERENCES – are limited to a total of 70 for Articles, Resources, Technical Reports; and 40 for Letters. This includes references in the main text and Methods combined. References must be numbered sequentially as they appear in the main text, tables and figure legends and Methods and must follow the precise style of Nature Cell Biology references. References only cited in the Methods should be numbered consecutively following the last reference cited in the main text. References only

associated with Supplementary Information (e.g. in supplementary legends) do not count toward the total reference limit and do not need to be cited in numerical continuity with references in the main text. Only published papers can be cited, and each publication cited should be included in the numbered reference list, which should include the manuscript titles. Footnotes are not permitted.

Methods should be written concisely, but should contain all elements necessary to allow interpretation and replication of the results. As a guideline, Methods sections typically do not exceed 3,000 words. The Methods should be divided into subsections listing reagents and techniques. When citing previous methods, accurate references should be provided and any alterations should be noted. Information must be provided about: antibody dilutions, company names, catalogue numbers and clone numbers for monoclonal antibodies; sequences of RNAi and cDNA probes/primers or company names and catalogue numbers if reagents are commercial; cell line names, sources and information on cell line identity and authentication. Animal studies and experiments involving human subjects must be reported in detail, identifying the committees approving the protocols. For studies involving human subjects/samples, a statement must be included confirming that informed consent was obtained. Statistical analyses and information on the reproducibility of experimental results should be provided in a section titled "Statistics and Reproducibility".

All Nature Cell Biology manuscripts submitted on or after March 21 2016 must include a Data availability statement at the end of the Methods section. For Springer Nature policies on data availability see <http://www.nature.com/authors/policies/availability.html>; for more information on this particular policy see <http://www.nature.com/authors/policies/data/data-availability-statements-data-citations.pdf>. The Data availability statement should include:

- Accession codes for primary datasets (generated during the study under consideration and designated as "primary accessions") and secondary datasets (published datasets reanalysed during the study under consideration, designated as "referenced accessions"). For primary accessions data should be made public to coincide with publication of the manuscript. A list of data types for which submission to community-endorsed public repositories is mandated (including sequence, structure, microarray, deep sequencing data) can be found here <http://www.nature.com/authors/policies/availability.html#data>.
- Unique identifiers (accession codes, DOIs or other unique persistent identifier) and hyperlinks for datasets deposited in an approved repository, but for which data deposition is not mandated (see here for details <http://www.nature.com/sdata/data-policies/repositories>).
- At a minimum, please include a statement confirming that all relevant data are available from the authors, and/or are included with the manuscript (e.g. as source data or supplementary information), listing which data are included (e.g. by figure panels and data types) and mentioning any restrictions on availability.
- If a dataset has a Digital Object Identifier (DOI) as its unique identifier, we strongly encourage including this in the Reference list and citing the dataset in the Methods.

We recommend that you upload the step-by-step protocols used in this manuscript to the Protocol Exchange. More details can be found at www.nature.com/protocolexchange/about.

All imaging data should be accompanied by scale bars, which should be defined in the legend. Cropped images of gels/blots are acceptable, but need to be accompanied by size markers, and to retain visible background signal within the linear range (i.e. should not be saturated). The boundaries of panels with low background have to be demarked with black lines. Splicing of panels should only be considered if unavoidable, and must be clearly marked on the figure, and noted in the legend with a statement on whether the samples were obtained and processed simultaneously. Quantitative comparisons between samples on different gels/blots are discouraged; if this is unavoidable, it should only be performed for samples derived from the same experiment with gels/blots were processed in parallel, which needs to be stated in the legend.

- We do not recommend using Adobe Photoshop for designing figures, but we can accept Photoshop generated (.PSD or .TIFF) files only if each element included in the figure (text, labels, pictures, graphs, arrows and scale bars) are on separate layers. All text should be editable in 'type layers' and line-art such as graphs and other simple schematics should be preserved and embedded within 'vector

smart objects' - not flattened raster/bitmap graphics.

The total number of Supplementary Figures (not including the “unprocessed scans” Supplementary Figure) should not exceed the number of main display items (figures and/or tables (see our Guide to Authors and March 2012 editorial <http://www.nature.com/ncb/authors/submit/index.html#suppinfo>; <http://www.nature.com/ncb/journal/v14/n3/index.html#ed>). No restrictions apply to Supplementary Tables or Videos, but we advise authors to be selective in including supplemental data.

GUIDELINES FOR EXPERIMENTAL AND STATISTICAL REPORTING

REPORTING REQUIREMENTS – To improve the quality of methods and statistics reporting in our papers we have recently revised the reporting checklist we introduced in 2013. We are now asking all life sciences authors to complete two items: an Editorial Policy Checklist (found here <https://www.nature.com/authors/policies/Policy.pdf>) that verifies compliance with all required editorial policies and a reporting summary (found here <https://www.nature.com/authors/policies/ReportingSummary.pdf>) that collects information on experimental design and reagents. These documents are available to referees to aid the evaluation of the manuscript. Please note that these forms are dynamic ‘smart pdfs’ and must therefore be downloaded and completed in Adobe Reader. We will then flatten them for ease of use by the reviewers. If you would like to reference the guidance text as you complete the template, please access these flattened versions at <http://www.nature.com/authors/policies/availability.html>.

We strongly recommend the presentation of source data for graphical and statistical analyses as a separate Supplementary Table, and request that source data for all independent repeats are provided when representative experiments of multiple independent repeats, or averages of two independent experiments are presented. This supplementary table should be in Excel format, with data for different figures provided as different sheets within a single Excel file. It should be labelled and numbered as

one of the supplementary tables, titled "Statistics Source Data", and mentioned in all relevant figure legends.

Author Rebuttal to Initial comments

Reviewer 1 comments (blue) and author responses (black):

The paper regarding Spatial heterogeneity of bone marrow endothelial cells unveils a distinct capillary subtype in the epiphysis is well presented. The authors showed that epiphysis consists of a spatial heterogeneity of BMECs, with features of capillary endothelial cells (ECs), termed type S (secondary ossification) ECs. They carried out genetic KO of VEGFR and Dll4 to investigate the bone phenotype and angiogenic features. They showed that type S ECs produce type I collagen, and module bone and hematopoietic development. This is an interesting study.

We sincerely appreciate the Reviewer's expert opinion and evaluation of the strength of our findings. According to the Reviewer's recommendations, we conducted additional experiments and revised the manuscript extensively, as described in the point-by-point responses provided below.

Are the cell surface markers of type S ECs overlapped and exchangeable to H type ECs during development?

Thank you for this important suggestion. ECs are highly plastic in principle, and their fate is not genetically pre-determined. For example, tip/stalk cells can become arterial or venous ECs (Luo et al., *Nature*, 2021). Indeed, some surface markers such as *Ly6a* and *Ly6c1* seemed to overlap in type H and type S ECs (**Fig. 1f**), suggesting these cells are similar. However, *Colla1* and *Colla2* were specific markers of type S ECs and were not expressed in type H ECs (**Fig. 1f, g**). Conversely, markers of type H ECs such as *Sox17* and *Efnb2* were not expressed in type S ECs (**Fig. 1e**). In the earlier stage at P3 when type S ECs were not yet present, immunohistochemical Ly6a expression was hardly visible in type H ECs (**Figures for reviewers 1a below**), and type H ECs were almost negative for *Colla1* and *Colla2* expression (**Figures for reviewers 1b below**). In addition, the epiphysial vasculature had no anatomically connection with the metaphyseal vasculature (**Fig. 2c, e, m**), suggesting that type S ECs are not derived from metaphyseal ECs. Nevertheless, we carefully specified this as “These data demonstrate that type S ECs possess unique phenotypic characteristics, although some of them overlap with those of type H ECs and AECs.” in the text (**p. 6, lines 127–128**).

Figures for reviewers 1

Figure 2 i. Can you provide some quantitative measurement instead of single %, how many times were done? N=?

We suspect you meant Figure 1i, not Figure 2i, in this comment. According to your suggestion, we quantified Ly6a^{hi}Ly6c1^{hi} cells in four independent experiments and provided the *p*-value in the corresponding figure legend (**new Fig. 1i**).

Figure 2, H-P. Can you provide some quantitative measurement for these described features?

According to the Reviewer's recommendation, we quantified the number of tip cell filopodia (**new Supplementary Fig. 2a**). The data indicate that filopodia are significantly abundant both in the epiphysis and metaphysis compared with the diaphysis, which possibly reflects the growth rate of the developing vasculature in each portion. We also quantified the relative intensities of Vegfr1-RFP, Vegfr2-GFP, and Vegfr3 (**new Supplementary Fig. 2b–d**). The data are largely consistent with the representative images shown in **Fig. 2o**.

Figure 3, There are three main subtypes of VEGFR, numbered 1, 2 and 3. Is VEGFR2 mainly expressed in epiphysis, relative to VEGFR1 and 3? What are the proportion of VEGFR1, 2, and 3 expression in epiphysis and metaphysis? What might happen if VEGFR1 and 3 were deleted?

Thank you for this suggestion. In terms of the images of reporter mice, expression of Vegfr2 and Vegfr3 did not apparently differ between type S and type H ECs (**Fig. 2o**;

new Supplementary Fig. 2b–d). They were only expressed in tip and stalk cells located in the growing edges of both the epiphysis and metaphysis. What makes these two subtypes different is that *Vegfr1* was highly expressed in type H ECs, but was undetectable in type S ECs (**Fig. 2o**). The lack of *Vegfr1*, a negative regulator buffering VEGF signaling (Potente et al., 2011), led us to hypothesize that type S ECs are more vulnerable to deletion of *Vegfr2*, the main signaling receptor for VEGF, than type H ECs. The purpose of the experiment shown in **Fig. 3** was to compare the vascular plasticity and responsiveness to VEGF between each EC subtype; therefore, we primarily deleted *Vegfr2* in ECs. The results showed that type S vessels are highly plastic during postnatal development, even more plastic than type H vessels, and robustly contribute to epiphyseal osteogenesis.

We preliminarily conducted knockout of *Vegfr1* in ECs (*Cdh5-Cre^{ERT2}xVegfr1-flox:Vegfr1^{ΔEC}*). Mutant mice showed defects in type H ECs but not in type S ECs (**Figures for reviewers 2 below**), which reflects its specific expression in type H ECs (**Fig. 2o**). In our view, the main focus of the current paper is to describe the characteristics of type S ECs, but not type H ECs; therefore, we chose not to show these data in order to retain the focus of the paper. Deletion of *Vegfr3* in ECs profoundly disrupts lymphatic vessels but minimally affects angiogenesis (Zarkada et al., 2015; Potente et al., 2011). Therefore, we did not perform *Vegfr3* deletion in the bone vasculature considering the purpose of this experiment stated above.

Figures for reviewers 2

Figure 3, There are four main subtypes of Delta-like protein, numbered 1, 2, 3 and 4. Why was Dll4 used to be chosen to delete? Is Dll4 mainly expressed in epiphysis,

relative to other subtypes? What are the proportion of Dll1-4 expression in epiphysis and metaphysis? What might happen if Dll like Dll1-3 were deleted?

In our scRNA-seq data, *Dll4* expression was largely ubiquitous in ECs and was not specific to type S ECs. However, considering the purpose of this figure is to compare the responsiveness of each EC subtype to angiogenic and antiangiogenic stimuli, we employed Dll4, the predominant Notch ligand in ECs that induces hypervascularity when its gene is deleted in ECs (Hellstrom et al., *Nature*, 2007; Benedito et al., *Nature*, 2012; Ramasamy et al., *Nature*, 2014). In these studies, *Dll4*-knockout mice were analyzed as the counterpart of *Vegfr2*-knockout mice; therefore, we followed this example. Examination of bone vasculature in mice lacking Dll1-3 is important considering Notch biology, and we would like to perform this in future experiments.

It was mentioned that type S ECs atypically contributed to the acquisition of bone strength by secreting type I collagen. Recent studies (for example, PMID: 32483430) have found that endothelial cells produce many angiocrine factors to regulate bone and cartilage via versatile mechanisms, including Hedgehog, Notch, WNT, bone morphogenetic protein (BMP), fibroblast growth factor (FGF), insulin-like growth factor (IGF), and platelet-derived growth factor (PDGF). It would be relevant to discuss this possibility by referring to this study. It would be informative to discuss if type S ECs might also produce other potential angiocrine factors with osteogenic property to regulate bone and cartilage, which would enhance the overview and potential of this study.

Thank you for this insightful suggestion. As you indicated, we suppose type I collagen is not the only angiocrine factor that regulates bone strength. The contribution of type S ECs to bone development must be versatile and supported by various other factors. Indeed, the phenotype of *Colla1*^{i^ΔEC} mice was less severe than that of *Vegfr2*^{i^ΔEC} mice (**Fig. 3, 4**). We clearly discussed this speculation in the paper as you indicated (**p. 14, lines 307–311**). According to the Reviewer's recommendation, we added all osteogenic angiocrine factors listed in the report suggested (Zhu et al., 2020 (PMID: 32483430)) to the heatmap (**new Fig. 4a**). We would like to highlight that expression of some such angiocrine genes, including *Il33*, *Wnt2*, *Shh*, and *Ihh*, was below the detectable level.

Figure 4 d-q, can you provide some quantitative measurement for these described features?

According to the Reviewer's recommendation, we quantified Osx^+ osteogenic progenitors in control and *Colla1^{ΔEC}* mice (**new Fig. 4p**).

Figure 5 a-g, can you provide some quantitative measurement for these described features?

According to the Reviewer's recommendation, we added the number of $c\text{-Kit}^+$ cells in the epiphysis at various stages (**new Fig. 5n**).

Figure 5 k-p, can you provide some quantitative measurement for these described features?

According to the Reviewer's recommendation, we added the number of $Cxcl12\text{-DsRed}^+$ cells in the epiphysis (**new Fig. 5n**).

It is not sure if type S ECs were observed in humans? It would be informative to show where type S ECs expressed in human bone?

Thank you for this insightful comment. Although the epiphysial vasculature is clearly separated from the metaphyseal vasculature in humans (please see the image below (Seeley et al., *Am Acad Ortho Surg*, 2016)), spatial and genetic diversity of BMECs has not been studied as far as we know. It is important to determine whether type S ECs are present in humans because epiphysial blood vessels are critically involved in human diseases such as Legg-Calve-Perthes disease (LCPD), a type of femoral head necrosis that affects children. There is no known cause of impaired blood supply to the femoral head in LCPD. Although it is extremely hard to get human bone samples of healthy (non-cancer) or LCPD children, we are keen to study this in the future. These points are now clearly discussed on **p. 14, lines 318–321**.

Figure 2

Seeley et al., Figure 2

Reviewer 2 comments (blue) and author responses (black):

In this study, the authors profiled bone marrow endothelial cells (ECs) isolated from mouse femurs, carefully segregating epiphysis, diaphysis and metaphysis regions of the mouse femur. Using scRNA-seq they identified a previously unappreciated capillary subtype, within the epiphysis termed type S ECs. These ECs manifested unique phenotypic signature Ly6ahighLy6chigh and expressed Col1a1 and Col1a2 which are commonly expressed by the non-vascular cells. The authors also demonstrated that repopulating HSPCs reside within epiphysis as well. Indeed, Type S ECs supported hematopoiesis by presumably secreting Cxcl12, Tgfb1, Tgfb2, and Spp1 (Osteopontin). They also used genetic models in which the expression of Col1a1, VEGFR2, and Dll4 among others were deleted in the adult endothelial cells. These studies demonstrated that Type S cells are delegated with supporting/balancing hematopoiesis and osteogenesis promoting bone growth/remodeling.

The contribution of the vessels in the epiphysis has been neglected as this portion of the bone was often discarded. Therefore, the angiocrine contribution of these unique endothelial cells (ECs) to balance of bone formation and hematopoiesis was not well studied. In this regard, the data presented in this paper that assigns an operational, phenotypic and functional unique attributes to Type S epiphyseal vessels are novel and could have both mechanistic and translational value. The finding that Type S epiphyseal ECs could deposit Col1a1 and Col1a2 as well as supply angiocrine factors Cxcl12, Tgfb1, Tgfb2 and osteopontin are based on genetic knock down of these factors specifically in the adult ECs and as such are convincing. Overall, the findings by the authors that unique population of ECs, namely Type S dendritic shaped vessels arborize the epiphyses is novel and can open the door for therapeutic approaches to study hematopoiesis and osteogenesis.

We appreciate the Reviewer's evaluation of the novelty and strength of our findings from the viewpoint of overall significance. We sincerely appreciate all the highly thoughtful and conceptual comments, which made us rethink what our data mean in the context of hematopoiesis and the stem cell system. According to these valuable comments, we revised the paper by including additional experiments or intensive discussion, while specifying the limitation of our study. Our point-by-point responses are provided below.

1) The demonstration that mesenchymal cell-derived type I collagen is insufficient for

epiphyseal osteogenesis, but that Type S EC-derived complement this insufficiency by secreting type I collagen is probably the most significant finding in this manuscript. This is important because to date no other tissue-specific ECs have been shown to express *Colla1,2* type collagens and the majority of the collagens produced by ECs are Collagen Type IV or other types of collagen. To prove this point, the authors have generated endothelial-specific *Colla1*-knockout (*Cdh5-BAC-CreERT2+Colla1flox/flox*, referred to as *Colla1 Δ EC*) mice, that manifested impaired osteogenesis. To further substantiate this important finding, beyond shadow of doubt, the authors might want to purify Type S ECs and show how these cells can synthesize *Colla1* or 2 as well as Collagen type IV or other types as well. The mechanism by which Type S ECs could produce *Colla1* or 2 and balance production of other collagen types should also be investigated.

We appreciate this valuable comment. The unique expression and function of type I collagen genes in type S ECs is one of the most surprising findings of the current study. As a confirmation, we provided the expression profile of other major collagen genes in each EC subtype (**new Fig. 1h**). The data show that expression of these genes is diverse among EC subtypes. Notably, only *Colla1* and *Colla2* are specifically expressed in type S ECs. Although we showed that *Colla1 Δ EC* mice had reduced collagen fibrils around blood vessels (**Fig. 4d-i**) and reduced bone strength (**Fig. 4q-t**), it is important to elucidate the precise mechanism by which type S ECs preferentially produce type I collagen over other collagens and what makes type S ECs distinct from other EC subtypes. Epigenetic regulation might account for this unique expression in type S ECs. Moreover, type S ECs might have characteristic features in multiple steps of collagen synthesis, including post-translational modification in the endoplasmic reticulum, release from the ribosome, glycosylation of procollagens, triple helix formation, and secretion (Claeys et al., *Hum Genet*, 2021). Examination of these intracellular events requires extensive biochemical experiments using purified type S ECs. However, the number of type S ECs we can harvest from mice is extremely small compared with the number required for such biochemical studies. We recognize the limitation of our study in terms of this important point and it is now carefully discussed on **p. 14, lines 312-317**.

2) Can the authors provide a schema in which Type S ECs sense hematopoietic mass and as such balance osteogenic activity as well? For example, is this balance is

regulated by differential activation of osteoblasts or osteoclasts? What is thermostat within the Type S ECs or their microenvironment that balance hematopoiesis and osteogenesis?

The bone marrow stem cell niche including BMECs acts as a biological rheostat for regulation of the hematopoietic system. Both humoral and cellular crosstalk is important for adaptation to changing demands for blood cells. In general, BMECs contribute to not just hematopoiesis but also osteogenesis. In this context, type S ECs may prioritize the latter compared with BMECs in other fragments, considering the small size of epiphyseal BM as a stem cell niche. Our transplantation analysis showed that type S ECs in epiphyses harbor *bona fide* HSCs, but systemic hematopoiesis at steady state should be maintained even if epiphyses are absent. Epiphyses likely act as a satellite niche, which might be utilized in an emergency such as extensive loss of the main BM cavity. It is unclear which biological factor regulates the prioritization of osteogenesis by type S ECs, but a high mechanical load placed on the metaphysis might be involved in the inclination of type S ECs to prioritize osteogenesis. Considering physical activity, type S ECs are critical because epiphyses sustain the stability of joints. This important point is now discussed on **p. 13, lines 289–297**.

In terms of the number of osteoclasts and osteoblasts, which are associated with BMECs, we did not find a difference between type S and type H ECs (**new Supplementary Fig. 4a; Fig. 3e, g**). Considering the reduction of mesenchymal/osteogenic progenitors in *Vegfr2^{ΔEC}* and *Dll4^{ΔEC}* mice (**Fig. 3e–k, p–v**), differentiation or proliferation, but not activation, of these cells is the major effect of type S ECs.

3) In figure 1, one can observe that as compared to Type L and Type H ECs, the number of Type S ECs in cluster 8 compose of a diminishingly a very small percentage of total ECs. In this regard, there could be under representation or drop out of certain transcriptomes in these scant population of Type S ECs. In addition, the FKPM values for *Colla1* and *Cola2* as well as *Cxcl12*, *Tgfb1*, *Tgfb2* and osteopontin is not well defined. If possible, the authors might consider performing bulk RNA sequencing to acquire a better understanding of the level of expression of these factors in Type S ECs.

We rechecked the raw UMI (Unique Molecular Identifier) counts and normalized expression of *Colla1* and *Colla2*. Both ridge and scatter plots (**new Supplementary Fig. 1d**) confirmed that definite populations of type S ECs express these genes

abundantly. Moreover, although the raw counts of these genes in the majority of type L and type S ECs were nearly zero, small populations of these ECs had substantial expression, suggesting that the possibility of dropout or underestimation is low. Single-cell analysis using the 10xGenomics system evaluates gene expression by the barcodes and UMI, but not FKPM (fragments per kilobase of exon per million reads mapped), for the purpose of adjustment of PCR bias and normalization. Deduplication by barcodes and UMI does not require adjustment by the gene length like FKPM. As is the case for FKPM, the UMI process inevitably cuts off low expression as you anticipated. However, the main claim of this study focuses on highly expressing genes and the analysis is always based on comparison of three EC subtypes. Therefore, the magnitude of the correlation for each gene is hardly affected by this general normalization process.

4) Type S ECs have low levels of Kit-ligand but high levels of Tgfb β s that might indicate this specific vascular niche is primarily delegated with supporting the HSPCs that are quiescent rather than undergoing self-renewal or expansion. While the authors show repopulating HSPCs reside in vicinity of the Type S ECs, the unique attributes of these HSPCs are unclear. For example, do Type S ECs prevent excessive differentiation of the HSPCs to osteoclastic cells? Do HSPCs could be mobilized from the Type S zoned region to the peripheral circulation?

We appreciate this insightful comment. As you noted, the expression profile of HSC/HSPC niche factors indicates that type S ECs preferentially support relatively quiescent (non-dividing) HSCs compared with other BMEC subtypes. To address this hypothesis, we evaluated the cell cycle state of HSPCs and HSCs (**new Fig. 6h, i**), and compared it between epiphyses and metaphyses. This showed that more HSPCs and HSCs were in G0 phase in epiphyses than in metaphyses, suggesting that epiphyseal HSCs are relatively dormant.

To visualize HSCs in the BM section, we performed CD150 staining employed in previous reports (Kunisaki et al. *Nature*, 2013; Kiel et al., *Cell*, 2005; Wang et al., *EMBO J*, 2013) (**new Supplementary Fig. 5h, i**). CD48 staining never colocalized with CD150 staining in our hands, and megakaryocytes could be discriminated by their morphology and positivity for CD31 (asterisks); therefore, we modified the original protocol and showed only CD150 staining. This modification aims to avoid non-canonical usage of multiple antibodies (CD41/CD48/Lin) assigning the same color to multiple antigens. The results showed that the distance between HSCs and ECs was

smaller in epiphyses than in metaphyses, suggesting that type S ECs preferentially harbor HSCs compared with type H ECs. Mobilization of HSPCs from the type S zoned region to the peripheral circulation could be tested by specifically marking HSCs/HSPCs in epiphyses. However, this cannot be performed using the current technology or existing genetic tools; therefore, we would like to perform it in future studies.

Reviewer 3 comments (blue) and author responses (black):

In this manuscript by Iga et al., the authors describe a subtype of bone marrow endothelial cells, termed type S ECs, in the epiphysis, which they demonstrate to have a role in regulating local ossification and providing a niche for HSCs. There are several interesting findings in the manuscript, but I have concerns about some of the analyses performed, particularly as relates to the characterization of the hematopoietic effects.

We sincerely appreciate the Reviewer's appraisal of the importance of our paper, and extremely insightful and constructive feedback, especially regarding the hematopoietic analyses. According to the Reviewer's comments, we performed additional experiments and modified our overstatement of the results. Please see our point-by-point responses to the Reviewer's comments below.

Figure 1. In Figure 1C, it appears in the UMAP that there is potential overlap between the "type S" ECs and the arteriole ECs. Also the arteriole ECs have high expression of *Ly6a*, which is noted to be high in the Type S ECs. Are the newly described type S ECs overlapping or redundant to the arteriolar ECs that others have previously characterized?

We apologize for this confusion. In **Fig. 1c**, cells overlapping with type S ECs (cluster 8) should be type L (cluster 4) colored deep green. As you commented, *Ly6a* seemed to overlap in arteriolar and type S ECs (**Fig. 1d**), suggesting these cells are similar. However, some arteriolar markers such as *Efnb2* and *Jag2* were barely detectable in type S ECs, at least based on the feature plots (**Figures for reviewers 3 below**). Conversely, *Colla1* and *Colla2*, which were specific to type S ECs, were not expressed in arteriolar ECs (**Fig. 1f, g**). Importantly, type S ECs and arteriolar ECs were clustered separately in the UMAP plot (**Fig. 1c**), reflecting a significant difference in their gross gene expression profiles. These data suggest that type S ECs are distinct from arteriolar ECs. Nevertheless, we carefully specified this as "These data demonstrate that type S ECs possess unique phenotypic characteristics, although some of them overlap with those of type H ECs and AECs." in the text (**p. 6, lines 127–128**).

Figures for reviewers 3

Figure 2. The images in Figure 2O are qualitatively interesting, but would be improved if expression of the VEGFR receptors was quantified. Authors should be able to perform flow cytometry on ECs from the different regions to quantify the different VEGFR isoforms on the different EC subsets.

According to the Reviewer’s recommendation, we quantified the number of tip cell filopodia (**new Supplementary Fig. 2a**). The data indicate that filopodia are significantly abundant in the epiphysis and metaphysis compared with the diaphysis, which possibly reflects the growth rate of the vasculature in each portion. We also quantified the immunoreactivities of Vegfr1-RFP, Vegfr2-GFP, and Vegfr3 (**new Supplementary Fig. 2b–d**). The values largely correlate with the representative images shown in **Fig. 2o**. Regarding flow cytometry, at least for Vegfr1, the antibody frequently used (R&D) does not work well in our hands. When we tested ECs from Vegfr1-CKO mice, massive non-specific staining was observed, raising doubts about the validity of this antibody.

The labeling in panel 2O is confusing. Any abbreviations shown in the figure panels should be spelled out in the Figure legend for that figure panel.

Thank you for pointing this out. We spelled out all abbreviations in the figure legends.

Figure 3. This panel elegantly displays the effects of inducible, EC-specific deletion of Vegfr2 or Dll4 on bone formation in the epiphysis and metaphysis via detection using Alizarin red and Osx+ cells. However, it would help to see the primary effects in high power on the epiphyseal and metaphyseal blood vessels at the same time points. Can the authors utilize additional EC markers to illuminate the vasculature at the same time

points in each group and show in higher power? The low power views of endomucin+ vessels/vascular structures in the epiphysis in panel 3M are not satisfactory.

We appreciate this important point. According to your comment, we provided high magnification views of blood vessels with additional EC markers, Erg and CD31 (**new Supplementary Fig. 2e, f**). The images are at the same time points. From these images, the morphology and EC marker expression of ECs that survived after deletion of *Vegfr2* and *Dll4* were not obviously impaired, but these ECs irregularly aggregated in epiphyses of *Vegfr2^{iΔEC}* and *Dll4^{iΔEC}* mice.

Figure 4. The differences in controls and the *Colla1*^{ΔEC} are subtle and not readily discerned by the images shown in panels l vs. m, n vs. o and p vs. q. It would be helpful to show higher magnification views to demonstrate the differences the authors are trying to show. Otherwise, the text should be qualified to reflect that the differences in these mice are modest and conclusions should be only carefully drawn from these data.

As you indicated, the numbers of *Osx*⁺ and *Runx2*⁺ cells did not apparently differ between control and *Colla1^{iΔEC}* mice (**former Fig. 4l–q**). We presented higher magnification views of these images (**new Fig. 4l–o**) and performed quantification. This demonstrated that there was no significant difference in *Osx*⁺ cells (**new Fig. 4p**). We interpreted the data as “Interestingly, mineral deposition evaluated by Alizarin red staining was significantly reduced in the epiphysis, but not in other bone portions, of *Colla1^{iΔEC}* mice (**Fig. 4j, k**). The numbers of pre-osteoprogenitors and osteoprogenitors were not significantly affected in *Colla1^{iΔEC}* mice (**Fig. 4l–p**); therefore, insufficient collagen frameworks might impair mineral deposition or stability (Reznikov et al., 2016)” (**p. 9, lines 196-200**).

Figure 5. Figure 5a-g: *c-kit*⁺ expression by itself is suboptimal to identify hematopoietic stem/progenitor cells. If the authors wish to highlight the anatomic association of hematopoietic stem/progenitor cells with vessels in the post natal epiphysis, then additional markers should be added, to include Sca-1 and additional lineage markers so that at least *ckit*⁺*sca-1*⁺*lin*⁻ cells can be identified in the epiphyses. Several labs have demonstrated staining for hematopoietic stem cells in situ using various markers, such as CD150⁺CD48/41⁻ (Kunisaki et al. Nature 2013;502:637) and the authors could follow any of these approaches to provide more convincing data regarding hematopoietic stem/progenitor association with vessels in the epiphysis post-natally.

We appreciate this important point. Although c-kit immunostaining worked well (**Fig. 5a–g, p, q, t**), Sca1 staining only marked ECs (**Fig. 1j**). Although this staining pattern is contradictory to well-known criteria in flow cytometric analysis of HSPCs, images of other groups (Kusumbe et al., *Nature*, 2016; Tikhonova et al., *Nature*, 2019) are consistent with our data. We think the flow cytometric data showing the existence of $ckit^+sca-1^+lin^-$ cells in the epiphysis (**Fig. 6a**) back up the histological data. According to your recommendation, we followed the CD150 staining employed by Kunisaki et al. (**new Supplementary Fig. 5h, i**). CD48 staining never colocalized with CD150 staining in our hands, and megakaryocytes could be discriminated by their morphology and positivity for CD31 (asterisks); therefore, we modified the original protocol and showed only CD150 staining. This modification aims to avoid non-canonical usage of multiple antibodies (CD41/CD48/Lin) assigning the same color to multiple antigens. The results showed that the distance between HSCs and ECs was smaller in epiphyses than in metaphyses.

The same concern noted above applies for panels 5k–p. If the authors are solely going to utilize $ckit^+$ to identify hematopoietic progenitors, then the text needs to be edited to reflect that limitation.

We appreciate this important point. Sca1 staining only marked ECs in bone histology, different from flow cytometric analysis; therefore, we toned down and edited the text as “Although Sca1 staining only marked endothelial cells (**Fig. 1j**), visualization of putative HSPCs by c-kit immunostaining (**Fig. 5a–g**) showed that these cells started to appear in the entry point of type S vessels around P11 (**Fig. 5d, e**)” (**p. 10, lines 211–214**).

Figure 5x. Please show representative flow cytometric plot of the 34-LSK cells in the control and *Vegfr2*^{ΔEC} EC group.

Thank you for this suggestion. We showed the CD34 plots for control and *Vegfr2*^{ΔEC} mice (**new Fig. 5w**).

Figure 6. Representative percentages of cell populations should be shown for each cell population gate shown in 6a.

According to your recommendation, we showed representative percentages of cell populations in **Fig. 6a**.

6b. If the percentages of phenotypic HSCs or long term HSCs are not different in the epiphysis versus the metaphysis, but the authors have observed a difference in HSC repopulation in transplanted mice, does this mean that the differences in donor cell repopulation in recipient mice are due to non-HSC effects, i.e. adventitial or supportive cells in the BM grafts? Were these transplants competitive? If not, they should be repeated using competitor cells to rigorously compare the HSC repopulating capacity of the donor hematopoietic cells from the epiphysis vs. the metaphysis.

One way to resolve the concern regarding non-HSC effects causing the differences in donor cell engraftment in recipient mice would be to FACS sort HSCs (i.e. CD150⁺CD48⁻lin⁻ BM cells) and transplant at limiting dilution, along with competitor BM cells into recipient mice since this would isolate comparison to the HSC populations in the donor mice.

As you noted, the percentage of phenotypic HSCs did not differ between the epiphysis and metaphysis (**Fig. 6a, b**); however, we observed a difference in HSC repopulation in transplanted mice (**Fig. 6c–g**). To resolve this discrepancy, we sorted CD150⁺CD48⁻ HSCs and transplanted them with limiting dilution, together with competitor BM cells, into recipient mice. The reconstitution activity of purified HSCs derived from the epiphysis was about 2.5 times higher than that of purified HSCs derived from the metaphysis (**new Supplementary Figure 6**), although the difference was not significant ($p = 0.19$, $n = 8$). This difference in the characteristic of HSCs in the transplantation assay could be ascribed to the lower division rate of epiphyseal HSCs than of metaphyseal HSCs as described later (**new Fig. 6h, i**).

Figure 6f. Does the diminution in donor engraftment in the epiphyseal donor BM cell – transplanted secondary mice reflect that both the epiphysis and the metaphysis contain relatively few long term HSCs compared to the diaphyseal portion of the mouse femur? (Acar et al. Nature 2015;526:126).

It has been suggested that quiescent HSCs reside in relative proximity to certain types of bone marrow ECs (e.g. arterioles), whereas more proliferative HSCs reside in different EC niches (Kunisaki et al. Nature 2013;502:637). Have the authors interrogated the cell cycle status of HSCs residing close to the type S epiphyseal ECs/vessels? This would be important information to add to the manuscript.

We appreciate this insightful comment. As you commented, in terms of the vascular niche, it is complicated and controversial where quiescent HSCs preferentially reside (e.g., arterial or sinusoidal). Kunisaki et al. reported that quiescent HSCs are predominantly present near arterial ECs, whereas Acar et al. reported that they mainly exist in the perisinusoidal area. Such controversy might be ascribed to a technical difference in BM imaging or use of different reporter mouse lines to identify HSCs. In the current study, such arterial and sinusoidal (Stab2⁺) ECs were very scarce in the epiphysis (**Fig. 1b, c, j**). Accordingly, the decline in engraftment of epiphyseal and metaphyseal donor BM cells in transplanted secondary mice (**Fig. 6f**) suggested that they contain relatively few LT-HSCs. In response to your comment, we evaluated the cell cycle state of HSPCs and HSCs (**new Fig. 6h, i**) and compared it between epiphyses and metaphyses. More HSPCs and HSCs were in G0 phase in epiphyses than in metaphyses, suggesting that epiphyseal HSCs are relatively dormant compared with metaphyseal HSCs. We discussed this on **p. 11, line 255- p.12 line 265**.

Decision Letter, first revision:

Our ref: NCB-A49914A

22nd June 2023

Dear Yoshiaki,

Thank you for submitting your revised manuscript "Spatial heterogeneity of bone marrow endothelial cells unveils a distinct subtype in the epiphysis" (NCB-A49914A). It has now been seen by the original referees and their comments are below. The reviewers find that the paper has improved in revision, and therefore we'll be happy in principle to publish it in Nature Cell Biology, pending minor revisions to comply with our editorial and formatting guidelines.

If the current version of your manuscript is in a PDF format, please email us a copy of the file in an editable format (Microsoft Word or LaTeX)-- we cannot proceed with PDFs at this stage.

Thank you again for your interest in Nature Cell Biology. Please do not hesitate to contact me if you have any questions.

Best wishes,
Stelios

Stylianos Lefkopoulos, PhD
He/him/his
Associate Editor
Nature Cell Biology
Springer Nature
Heidelberger Platz 3, 14197 Berlin, Germany

E-mail: stylianos.lefkopoulos@springernature.com

Twitter: @s_lefkopoulos

Reviewer #1 (Remarks to the Author):

This is a revised paper. The authors have addressed questions.

Reviewer #2 (Remarks to the Author):

The authors have performed extensive additional experiments and provided clear explanations for the majority of my major concerns.

They now show more convincingly the unique phenotypic and functional features of previously unappreciated epiphyseal type S endothelial cells. They show the differential pattern of the VEGF-A and VEGF-C receptor expression by these type S vessels.

They also provide more cogent evidence for the expression of Col1a1 and Col1a2 by Type S endothelial cells.

While it is still not clear whether Type S vessels host true repopulating HSCs, they show that there is a trend in the localization of the primitive HSCs to the epiphyseal type S vessels.

If the authors could have documented the presence of type S vessels in the human epiphyseal bones, the novelty of this paper would have increased. But this referee acknowledges the difficulty in acquiring intact human bones.

Overall, the revised manuscript is significantly improved.

Reviewer #3 (Remarks to the Author):

The authors have satisfactorily addressed most of my concerns and have performed the necessary additional experiments to strengthen their conclusions.

Decision Letter, Final Checks:

Our ref: NCB-A49914A

6th July 2023

Dear Dr. Kubota,

Thank you for your patience as we've prepared the guidelines for final submission of your Nature Cell Biology manuscript, "Spatial heterogeneity of bone marrow endothelial cells unveils a distinct subtype in the epiphysis" (NCB-A49914A). Please carefully follow the step-by-step instructions provided in the attached file, and add a response in each row of the table to indicate the changes that you have made. Please also check and comment on any additional marked-up edits we have proposed within the text. Ensuring that each point is addressed will help to ensure that your revised manuscript can be swiftly handed over to our production team.

In recognition of the time and expertise our reviewers provide to Nature Cell Biology's editorial process, we would like to formally acknowledge their contribution to the external peer review of your manuscript entitled "Spatial heterogeneity of bone marrow endothelial cells unveils a distinct subtype in the epiphysis". For those reviewers who give their assent, we will be publishing their names alongside the published article.

Nature Cell Biology offers a Transparent Peer Review option for new original research manuscripts submitted after December 1st, 2019. As part of this initiative, we encourage our authors to support increased transparency into the peer review process by agreeing to have the reviewer comments, author rebuttal letters, and editorial decision letters published as a Supplementary item. When you submit your final files please clearly state in your cover letter whether or not you would like to participate in this initiative. Please note that failure to state your preference will result in delays in accepting your manuscript for publication.

Cover suggestions

As you prepare your final files we encourage you to consider whether you have any images or illustrations that may be appropriate for use on the cover of Nature Cell Biology.

Nature Cell Biology has now transitioned to a unified Rights Collection system which will allow our Author Services team to quickly and easily collect the rights and permissions required to publish your work. Approximately 10 days after your paper is formally accepted, you will receive an email in providing you with a link to complete the grant of rights. If your paper is eligible for Open Access, our Author Services team will also be in touch regarding any additional information that may be required to arrange payment for your article.

Please note that *Nature Cell Biology* is a Transformative Journal (TJ). Authors may publish their research with us through the traditional subscription access route or make their paper immediately open access through payment of an article-processing charge (APC). Authors will not be required to

make a final decision about access to their article until it has been accepted. Find out more about Transformative Journals

Authors may need to take specific actions to achieve compliance with funder and institutional open access mandates. If your research is supported by a funder that requires immediate open access (e.g. according to Plan S principles) then you should select the gold OA route, and we will direct you to the compliant route where possible. For authors selecting the subscription publication route, the journal's standard licensing terms will need to be accepted, including self-archiving policies. Those licensing terms will supersede any other terms that the author or any third party may assert apply to any version of the manuscript.

Please use the following link for uploading these materials:
[Redacted]

Best regards,

Kendra Donahue
Staff
Nature Cell Biology

On behalf of

Stylios Lefkopoulos, PhD
He/him/his
Associate Editor
Nature Cell Biology
Springer Nature
Heidelberger Platz 3, 14197 Berlin, Germany

E-mail: stylios.lefkopoulos@springernature.com
Twitter: @s_lefkopoulos

Reviewer #1:
Remarks to the Author:

This is a revised paper. The authors have addressed questions.

Reviewer #2:

Remarks to the Author:

The authors have performed extensive additional experiments and provided clear explanations for the majority of my major concerns.

They now show more convincingly the unique phenotypic and functional features of previously unappreciated epiphyseal type S endothelial cells. They show the differential pattern of the VEGF-A and VEGF-C receptor expression by these type S vessels.

They also provide more cogent evidence for the expression of Col1a1 and Col1a2 by Type S endothelial cells.

While it is still not clear whether Type S vessels host true repopulating HSCs, they show that there is a trend in the localization of the primitive HSCs to the epiphyseal type S vessels.

If the authors could have documented the presence of type S vessels in the human epiphyseal bones, the novelty of this paper would have increased. But this referee acknowledges the difficulty in acquiring intact human bones.

Overall, the revised manuscript is significantly improved.

Reviewer #3:

Remarks to the Author:

The authors have satisfactorily addressed most of my concerns and have performed the necessary additional experiments to strengthen their conclusions.

Final Decision Letter:

Dear Yoshiaki,

I am pleased to inform you that your manuscript, "Spatial heterogeneity of bone marrow endothelial cells unveils a distinct subtype in the epiphysis", has now been accepted for publication in Nature Cell Biology. Congratulations!

Over the next few weeks, your paper will be copyedited to ensure that it conforms to Nature Cell

Biology style. Once your paper is typeset, you will receive an email with a link to choose the appropriate publishing options for your paper and our Author Services team will be in touch regarding any additional information that may be required.

Please note that *Nature Cell Biology* is a Transformative Journal (TJ). Authors may publish their research with us through the traditional subscription access route or make their paper immediately open access through payment of an article-processing charge (APC). Authors will not be required to make a final decision about access to their article until it has been accepted. Find out more about Transformative Journals

If you have not already done so, we strongly recommend that you upload the step-by-step protocols

used in this manuscript to the Protocol Exchange (www.nature.com/protocolexchange), an open online resource established by Nature Protocols that allows researchers to share their detailed experimental know-how. All uploaded protocols are made freely available, assigned DOIs for ease of citation and are fully searchable through nature.com. Protocols and Nature Portfolio journal papers in which they are used can be linked to one another, and this link is clearly and prominently visible in the online versions of both papers. Authors who performed the specific experiments can act as primary authors for the Protocol as they will be best placed to share the methodology details, but the Corresponding Author of the present research paper should be included as one of the authors. By uploading your Protocols to Protocol Exchange, you are enabling researchers to more readily reproduce or adapt the methodology you use, as well as increasing the visibility of your protocols and papers. You can also establish a dedicated page to collect your lab Protocols. Further information can be found at www.nature.com/protocolexchange/about

With kind regards,
Stelios

Stylianos Lefkopoulos, PhD
He/him/his
Associate Editor
Nature Cell Biology
Springer Nature
Heidelberger Platz 3, 14197 Berlin, Germany

E-mail: stylianos.lefkopoulos@springernature.com
Twitter: @s_lefkopoulos